# Language Models Linearly Represent Sentiment

**Curt Tigges[1,4], Oskar J. Hollinsworth[2,4], Atticus Geiger[4], Neel Nanda,**

[1]Decode Research, [2]FAR AI, [3]Pr(AI)[2]R, [4]SERI MATS,
**Correspondence:** ct@curttigges.com

## Abstract

Sentiment is a pervasive feature in natural language text, yet it is an open question how sentiment is represented within Large Language Models (LLMs). In this study, we reveal that across a range of models, sentiment is represented linearly: a single direction in activation space mostly captures the feature across a range of tasks with one extreme for positive and the other for negative. In a causal analysis, we isolate this direction using interventions and show it is causal in both toy tasks and real world datasets such as Stanford Sentiment Treebank.

We analyze the mechanisms that involve this direction and discover a phenomenon which we term the summarization motif: sentiment is not just represented on valenced words, but is also summarized at intermediate positions without inherent sentiment, such as punctuation and names. We show that in SST classification, ablating the sentiment direction across all tokens results in a drop in accuracy from 100% to 62% (vs. 50% random baseline), while ablating the summarized sentiment direction at comma positions alone produces close to half this result (reducing accuracy to 82%).

## 1 Introduction

Large language models (LLMs) have displayed increasingly impressive capabilities (Brown et al., 2020; Radford et al., 2019; Bubeck et al., 2023), but their internal workings remain poorly understood. Nevertheless, recent evidence (Li et al., 2023) has suggested that LLMs are capable of forming models of the world, i.e., inferring hidden variables of the data generation process rather than simply modeling surface word co-occurrence statistics. There is significant interest (Christiano et al. (2021), Burns et al. (2022)) in deciphering the latent structure of such representations.

In this work, we investigate how LLMs represent sentiment, a variable in the data generation process that is relevant and interesting across a wide variety of language tasks (Cui et al., 2023). Approaching our investigations through the frame of causal mediation analysis (Vig et al., 2020; Pearl, 2022), we show that these sentiment features are represented linearly by the models, are causally significant, and are utilized by human-interpretable circuits (Olah et al., 2020; Elhage et al., 2021a).

We find the existence of a single direction scientifically interesting as further evidence for the linear representation hypothesis (Mikolov et al., 2013; Elhage et al., 2022; Park et al., 2023; Jiang et al., 2024), that models tend to extract properties of the input and internally represent them as directions in activation space. Understanding the structure of internal representations is crucial to begin to decode them. Linear representations are particularly amenable to detailed reverse-engineering (Nanda et al., 2023b) and have seen recent interest in the context of Sparse Autoencoders (Bricken et al., 2023). We believe that interpreting internal representations in LLMs shows promise for mitigating problematic behaviours.

We show evidence of a phenomenon which we have labeled the "summarization motif"[1], where rather than sentiment being directly moved from valenced tokens to the final token, it is first aggregated on intermediate summarization tokens without inherent valence such as commas, periods and particular nouns. This can be seen as a naturally emerging analogue to the explicit classification token in BERT-like models (Devlin et al., 2018), and in that context the phenomenon was observed by Clark et al. (2019). We show that the sentiment stored on summarization tokens is causally relevant for the final prediction. We find this an intriguing example of an "information bottleneck", where the data generation process is funnelled through a small subset of tokens used as information stores.

---

[1]Crucially, this is not to be confused with the NLP summarization task

Understanding the existence and location of information bottlenecks is a key first step to deciphering world models. This finding additionally suggests the models' ability to create summaries at various levels of abstraction, in this case at a sentence or clause rather than a token.

Our contributions are as follows. In Section 3, we demonstrate that standard methods can find a **linear representation of sentiment** using a toy dataset, and show that this direction correlates with sentiment information in the wild. We use causal analysis to show that this linear representation matters causally in both toy and crowdsourced datasets. In Section 4, we show through activation patching (Geiger et al., 2020; Vig et al., 2020) and ablations (techniques defined in Section 2.2) that the learned sentiment direction is used in **summarization behavior** that is causally important to circuits performing sentiment tasks. We replicate these findings across GPT2, Pythia, Gemma, Qwen and StableLM models (Section 2.1). In sum, we provide a novel, detailed case study of how to analyse a feature's representation in activation space.

## 2 Methods

### 2.1 Datasets and Models

**ToyMovieReview**    is a templatic dataset of continuation prompts we generated with the form "I thought this movie was ADJECTIVE, I VERBed it. Conclusion: This movie is" where ADJECTIVE and VERB are either two positive words (e.g., incredible and enjoyed) or two negative words (e.g., horrible and hated) that are sampled from a fixed pool of 85 adjectives (split 55/30 for train/test) and 8 verbs. The expected completion for a positive review is one of a set of positive descriptors we selected from among the most common completions (e.g. great) and the expected completion for a negative review is a similar set of negative descriptors (e.g., terrible). This dataset is the simplest toy task we could imagine to elicit understanding of sentiment in the smallest models that we tested through a next-token prediction task, avoiding the need for fine-tuning.

**ToyMoodStory**    is a similar toy dataset which is multi-subject and character-driven with random names, e.g. Carl hates parties, and avoids them whenever possible. Jack loves parties, and joins them whenever possible. One day, they were invited to a grand gala. Jack feels very [excited/nervous]

**Stanford Sentiment Treebank (SST)**    (Socher et al., 2013) consists of 10,662 one sentence movie reviews with human annotated sentiment labels for every constituent phrase from every review.

**Internet Movie Database (IMDB)**    (Maas et al., 2011) consists of 25,000 movie reviews taken from the IMDB website with human-annotated sentiment labels for each review.

**OpenWebText**    (Gokaslan and Cohen, 2019) is the pretraining dataset for GPT-2 which we use as a source of random text for correlational evaluations.

**GPT-2 and Pythia**    (Radford et al., 2019; Biderman et al., 2023) are families of decoder-only transformer models with sizes varying from 85M to 2.8b parameters. We mostly focus on Pythia-2.8b in the main body of this paper, reducing to Pythia-1.4b or GPT2-small when appropriate for saving compute, and leaving demonstrations of consistency across models to A.6.4 and A.9.2.

### 2.2 Causal Analysis Methods

**Activation patching**    Activation patching (Geiger et al., 2020; Vig et al., 2020), we create two symmetrical datasets $X_{\text{orig}}$ and $X_{\text{flipped}}$, where each prompt $x_{\text{orig}}$ and its counterpart prompt $x_{\text{flipped}}$ are of the same length and format but where key words are changed in order to flip the sentiment; e.g., "This movie was great" could be paired with "This movie was terrible". Let $\mathbb{A}$ be the set of all hidden layer activations of the model. We first conduct baseline forward passes, capturing the tensors of all activation values $\mathbf{A}_{\text{orig}} = \mathcal{F}(x_{\text{orig}})$, $\mathbf{A}_{\text{flipped}} = \mathcal{F}(x_{\text{flipped}})$ for intermediate activations $\mathbb{A}$. We then conduct "patched" forward passes using $x_{\text{flipped}}$ as $\mathbf{A}_{\mathbb{C}} = \mathcal{F}(x_{\text{flipped}}, \mathbf{A}_{\text{orig}}, \mathbb{C})$ for different model components $\mathbb{C} \subset \mathbb{A}$ representing a subset of the activations, where at each intermediate computation $I(a)$ in the forward pass taking a member $i \in \mathbb{C}$ as an input, we substitute or "patch" the alternate activation $a \mapsto a_{\text{orig}} := \mathbf{A}_{\text{orig}}[i]$ and instead compute $I(a_{\text{orig}})$. We can thus determine the relative importance of various model components $\mathbb{C}$ with respect to the task currently being performed, using some task performance metric (options discussed in Section 2.3) $\mathcal{M} : \mathbf{A} \mapsto \mathbb{R}$.

**Directional activation patching**    Geiger et al. (2023b) introduce a variant of activation patching that we call "directional activation patching". The idea is that rather than modifying the standard basis

directions of a component, we instead only modify the component along a single direction in the vector space, represented by unit vector $\hat{d}$, replacing it during a forward pass with the value from a different input. That is, the "patch" becomes $\mathcal{M}_{\mathbb{C} \leftarrow \mathbf{C}_{\text{flipped}} - \mathbf{C}_{\text{flipped}} \cdot \hat{d} + \mathbf{C}_{\text{orig}} \cdot \hat{d}}(\boldsymbol{x}_{\text{flipped}})$.

**Freezing** To analyze how the causal effect of a component $\mathbb{C}$ is mediated by another component $\mathbb{D}$, we perform an activation patch on $\mathbb{C}$ while freezing the activations of $\mathbb{D}$ to their initial value from the forward pass on the flipped prompt. We perform a forward pass with the flipped input to obtain an intervened model state $\mathcal{M}_{\mathbb{C} \leftarrow \mathbf{C}_{\text{orig}}; \mathbb{D} \leftarrow \mathbf{D}_{\text{flipped}}}(\boldsymbol{x}_{\text{flipped}})$. In particular, we can run patching experiments with frozen attention, meaning that the attention pattern is frozen from the original run so that the model still weights the value vectors in the same way, helping to isolate V-composition.

**Ablations** To capture the importance of a component, we eliminate its contribution by replacing it with zeros (zero ablation) or the mean activation over some dataset (mean ablation). Like activation patching, ablation is an intervention on a model component. However, the intervened activations are all zeros or taken from the mean over some dataset rather than from a paired forward pass. i.e. $\mathcal{M}_{\mathbb{C} \leftarrow \mathbf{C}^{\text{ablation}}}(\boldsymbol{x})$ where $\mathbf{C}^{\text{ablation}}$ consists of all zeros or a mean value. We also perform directional ablation, in which a component's activations are ablated only along a specific direction.

## 2.3 Evaluation metrics

**Logit difference metric** We extend the logit difference metric used by Wang et al. (2022) to the setting with 2 *classes* of next token rather than only 2 valid next tokens. This is useful in situations where there are many possible choices of positively or negatively valenced next tokens.

Specifically, we examine the average difference in logits between sets of positive/negative next-tokens $T^{\text{pos}} = \{t_i^{\text{pos}} : 1 \leq i \leq n\}$ and $T^{\text{neg}} = \{t_i^{\text{neg}} : 1 \leq i \leq n\}$ in order to get a smooth measure of the model's ability to differentiate between sentiment. That is, we define the logit difference for input $x$ as $\frac{1}{n} \sum_i \left[ \text{logit}(\mathcal{M}(x); t_i^{\text{pos}}) - \text{logit}(\mathcal{M}(x); t_i^{\text{neg}}) \right]$. Larger differences indicate more robust separation of the positive/negative tokens, and zero or inverted differences indicate zero or inverted sentiment processing respectively. When used as a patching

metric, this shows the causal efficacy of various interventions like activation patching or ablation.

We use this metric often because it is more sensitive than accuracy to small shifts in model behavior, which is particularly useful for circuit identification where the effect size is small but real. That is, in many cases a token of interest might become much more likely but not cross the threshold to change accuracy metrics, and in this case logit difference will detect it. Logit difference is also useful when trying to measure the model behavior transition between two different, opposing prompts–in this case, the logit difference for each of the prompts is used for lower and upper baselines, and we can measure the degree to which the logit difference behavior moves from one pole to the other.

**Logit flip metric** We also extend the interchange intervention accuracy metric from Geiger et al. (2022) to classes of tokens by computing the percentage of cases where the logit difference between $T^{\text{positive}}$ and $T^{\text{negative}}$ is inverted after an intervention. This is a more discrete measure which is helpful for gauging whether the magnitude of the logit differences is sufficient to flip model predictions.

**Accuracy** Out of a set of prompts, the percentage for which the logits for tokens $T^{\text{correct}}$ are greater than $T^{\text{incorrect}}$. Usually each of these sets only has one member (e.g., "Positive" and "Negative").

## 2.4 Finding Directions

Here we defined three methods to find a sentiment direction in each layer of a language model using our ToyMovieReview dataset. In each of the following, let $\mathbb{P}$ be the set of positive inputs and $\mathbb{N}$ be the set of negative inputs. For some input $x \in \mathbb{P} \cup \mathbb{N}$, let $\boldsymbol{a}_x^L$ and $\boldsymbol{v}_x^L$ be the vector in the residual stream at layer $L$ above the adjective and verb respectively. We reserve $\{\boldsymbol{v}_x^L\}$ as a hold-out set for testing. Let the correct next token for $\mathbb{P}$ be $p$ and for $\mathbb{N}$ be $n$.

**$k$-means (KM)** We fit 2-means to $\{\boldsymbol{a}_x^L : x \in \mathbb{P} \cup \mathbb{N}\}$, obtaining cluster centroids $\{\boldsymbol{c}_i : i \in [0, 1]\}$ and take the direction $\boldsymbol{c}_1 - \boldsymbol{c}_0$.

**Linear Probing** The direction is the normed weights $\frac{\boldsymbol{w}}{||\boldsymbol{w}||}$ of a logistic regression (**LR**) classifier $\mathbf{LR}(a_x^L) = \frac{1}{1 + \exp(-\boldsymbol{w} \cdot \boldsymbol{a}_x^L)}$ trained to distinguish between $x \in \mathbb{P}$ and $x \in \mathbb{N}$.

**Distributed Alignment Search (DAS)** We perform directional patching (2.2), pairing up inputs $p \in \mathbb{P}, n \in \mathbb{N}$, then patching as $\boldsymbol{a}_p \mapsto \boldsymbol{a}_p - \boldsymbol{a}_p \cdot$

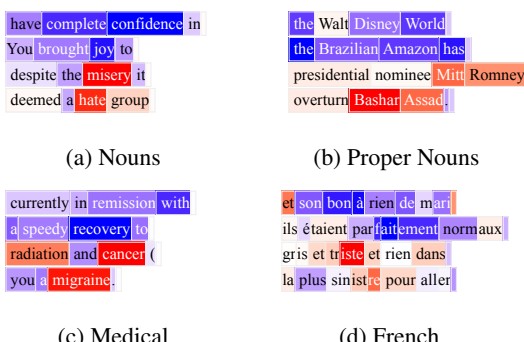

(a) Nouns      (b) Proper Nouns

(c) Medical      (d) French

Figure 1: Visualizing the "sentiment activation" (projection of the residual stream onto the sentiment axis) where blue is positive and red is negative. Examples (1a-1c) show the $k$-means sentiment direction for the first layer of GPT2-small on samples from OpenWebText. Example 1d shows the $k$-means sentiment direction for the 7th layer of Pythia-1.4b on the opening of Harry Potter in French.

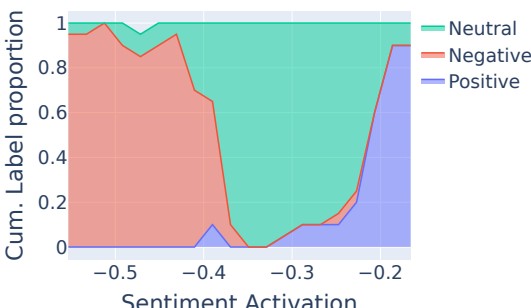

Figure 2: Area plot of sentiment labels for OpenWeb-Text samples by sentiment activation, i.e. the projection of the first residual stream layer of Pythia-2.8b at that token onto the sentiment direction. The sentiment activation acts as a strong classifier, separating positive and negative tokens from a real dataset. Ground truth classification was performed by GPT-4. Direction was fit using $k$-means.

$\boldsymbol{\theta} + \boldsymbol{a}_n \cdot \boldsymbol{\theta}$ (and vice versa). The patching metric is the logit difference

$$\mathcal{M}(\boldsymbol{\theta}) = \sum_{x \in \mathbb{P}} [\mathrm{logit}_\theta(x; p) - \mathrm{logit}_\theta(x; n)] + \sum_{x \in \mathbb{N}} [\mathrm{logit}_\theta(x; n) - \mathrm{logit}_\theta(x; p)] .$$

We then determine $\boldsymbol{\theta}$ as $\boldsymbol{\theta} = \arg\max_{\|\boldsymbol{\theta}\|=1} \mathcal{M}(\boldsymbol{\theta})$, which we approximate using gradient descent. This generalizes to finding a $k$-dimensional *subspace* by fitting an orthonormal rotation matrix $\boldsymbol{R}$ which maximizes $\mathcal{M}(\boldsymbol{R})$, patching only the first $k$ components in the rotated basis $\boldsymbol{a}_p \mapsto \boldsymbol{a}_p + \boldsymbol{R}^T ([\boldsymbol{R}(\boldsymbol{a}_n - \boldsymbol{a}_p)]_{i:i \leq k})$ and then the subspace is the span of the first $k$ rows of $\boldsymbol{R}$.

## 3 Finding a "Sentiment Direction"

The first question we investigate is whether there exists a direction in the residual stream in a transformer model that represents the sentiment of the input text, as a special case of the linear representation hypothesis (Mikolov et al., 2013; Park et al., 2023; Jiang et al., 2024), that features are represented linearly as directions in activation space. We show that the methods discussed above (e.g. $k$-means, LR and DAS, see Section 2.4) all arrive at a similar sentiment direction. We can visualize the feature being represented by this direction by projecting the residual stream at a given token/layer onto it, using some text from the training distribution. We will call this the "**sentiment activation**".

**Finding and Comparing the Directions** To find initial directions corresponding with sentiment, we first fit directions from the residual stream over the adjective token in the ToyMovieReview dataset (Section 2.1), using methods from Section 2.4. We find extremely high cosine similarity (Figure A.1) between the directions yielded by each of these methods in Pythia-2.8b (cf. A.7 for other models). This suggests that these are all noisy approximations of the same direction, and indeed our results appear robust to choice of fitting method.

### 3.1 Correlational Evaluation

To examine the relationship between the directions we had identified and real-world text, we investigated how these directions correlate with sentiment in natural text, as evaluated by human readers and advanced LLMs (GPT-4).

**Visualizing The Sentiment Direction** By way of making initial comparisons between the sentiment direction and real-world text, we show (Figure 1) a visualisation in the style of Neuroscope (Nanda, 2023b) where the sentiment activation (the projection of the residual stream onto the sentiment axis) is represented by color, with red being negative and blue being positive. It is important to note that the direction being examined here was produced by training on just 30 positive and 30 negative English adjectives in an unsupervised way (using $k$-means with $k = 2$). Notwithstanding, the extreme values along this direction appear readily interpretable in the wild, even in diverse text domains such as the

| direction | flip percent | flip median size |
|-----------|--------------|------------------|
| DAS | 96% | 107% |
| KM | 96% | 69% |
| LR | 100% | 86% |

Figure 3: We created a dataset of 27 negation examples and compute the change in $k$-means sentiment activation (projection of the residual stream onto the sentiment axis) at the negated token (e.g. "doubt") between the 1st and 10th resid-post layers of GPT2-small. Here "flip percent" is the percentage of the 27 prompts for which the sign of the sentiment activation has flipped and "flip median size" is the median size of the flip relative to the size of the initial sentiment activation.

Figure 4: Visualizing the sentiment activations across the layers of GPT2-small for a text where the sentiment hinges on negations. Color represents sentiment activation (projection of the residual stream onto the sentiment axis) at the given layer and position. Red is negative, blue is positive. Each row is a residual stream layer, first layer is at the top.

opening paragraphs of Harry Potter in French.

**Quantifying classification accuracy** To rigorously validate this visual check, we binned the sentiment activations of OpenWebText tokens from the first residual stream layer of GPT2-small into 20 equal-width buckets and sampled 20 tokens from each. Then we asked GPT-4 to classify into Positive/Neutral/Negative.[2] In Figure 2, we show an area plot of the classifications by activation bin in Pythia-2.8b (cf. Figure A.8 for other models). Defining a classifier using a threshold of the top/bottom 0.1% of sentiment activations in GPT2-small, we can achieve over 90% accuracy as compared to GPT-4 classifications as our ground truth (Figure A.8a). In the area plot we can see that the left side area is dominated by the "Negative" label, whereas the right side area is dominated by the "Positive" label and the central area is dominated by the "Neutral" label. Hence the tails of the activations seem highly interpretable as representing a bipolar sentiment feature. The large space in the middle of the distribution simply occupied by neutral words (rather than a more continuous degradation of positive/negative) indicates superposition of features (Elhage et al., 2022).

**Negation Flips the Sentiment Direction in Later Layers** Using the $k$-means sentiment direction after the first layer of GPT2-small, we can obtain a view of how the model updates its view of sentiment during the forward pass, analogous to the

"logit lens" technique from nostalgebraist (2020). The example text that we use here is "You never fail. Don't doubt it. I don't like you". In Figure 4, we see how the sentiment activation flips when the context of the sentiment word denotes that it is negated. The words 'fail' and 'doubt' can be seen to flip from negative in the first couple of layers to being positive after a few layers of processing. In contrast, the word 'like' flips from positive to negative. We quantified this result using a toy dataset of 27 similar examples and computed the flip in sentiment activation during the forward pass for different direction finding methods (Figure 3).

### 3.2 Causal Evaluation

The experiments described so far illustrate only correlations between our identified directions and sentiment. In order to demonstrate that these directions are indeed causal, we used causal mediation analysis on our toy dataset and validated our findings on two different real world datasets.

**Sentiment directions are causally active.** We evaluate the sentiment direction using directional patching on the adjective and verb token representations for each layer (Section 2.2) in Table 1. These evaluations are performed on prompts with out-of-sample adjectives and the direction was not trained on *any* verbs. We find that patching activations along a single direction can cause a significant change in the prediction according to both of our patching metrics, and the direction found using DAS is able to completely flip the prediction.

**Validation on SST** We validate our sentiment directions derived from toy datasets (Section 3.2) on SST. We collapsed the labels down to a binary "Pos-

---

[2]We gave GPT-4 prompts of the form: "Your job is to classify the sentiment of a given token (i.e. word or word fragment) into Positive/Somewhat positive/Neutral/Somewhat negative/Negative. Token: '{token}'. Context: '{context}'. Sentiment: " where the context length was 20 tokens centred around the sampled token.

| Method | ToyMovieReview | Treebank |
|---|---|---|
| DAS (1 dim.) | 109.8% | 47.0% |
| DAS (2 dim.) | 110.4% | 42.8% |
| DAS (3 dim.) | 110.2% | 35.9% |
| $k$-means | 67.2% | 22.1% |
| LR | 71.1% | 30.8% |
| Random | 0.4% | 0.1% |

(a) Logit difference metric: mean % change in logit difference (100% for one example means the sign of the logit difference has flipped while the magnitude is unchanged)

| Method | ToyMovieReview | Treebank |
|---|---|---|
| DAS (1 dim.) | 100.0% | 53.5% |
| DAS (2 dim.) | 95.5% | 49.0% |
| DAS (3 dim.) | 95.5% | 39.4% |
| $k$-means | 72.7% | 14.8% |
| LR | 86.4% | 16.8% |
| Random | 0.0% | 0.6% |

(b) Logit flip metric: the percentage of examples for which the logit difference changes sign

Table 1: Directional patching results for different methods in Pythia-1.4b (2.8b not shown due to compute time). We report the best result found across layers. The columns show two evaluation datasets, ToyMovieReview and Treebank. We present two different evaluation metrics in 1a and 1b.

itive"/"Negative", took the unique phrases from the source sentences, restricted to the 'test' partition and took a subset where Pythia-1.4b can achieve 100% zero-shot accuracy, removing 17% of examples. Then we paired up phrases of an equal number of tokens[3] to make up 460 clean/corrupted pairs. We used the scaffolding "Review Text: TEXT, Review Sentiment:" and evaluated the logit difference between "Positive" and "Negative" as our patching metric. Using the same DAS direction from Section 3 trained on just a few examples and flipping the corresponding sentiment activation between clean/corrupted in a single layer, we can flip the model's prediction 53.5% of the time (Table 1). The sentiment direction learned from a toy dataset can control behavior on a crowd-sourced dataset, which is a remarkable generalization result.

**Validation at the document level** In order to verify the applicability of our findings to larger document-sized prompts, we performed directional ablation (2.2) on the IMDB dataset, most of which consists of multiple sentences. Each item of this dataset was appended with "Review Sentiment:" in order to prompt a classification completion, and we selected 1000 examples each from the positive and negative items that the model was capable of classifying correctly. We used the sentiment directions found with DAS to ablate sentiment at every token at every layer (using Pythia-2.8b). As a result, classification accuracy dropped from 100% to 57%, suggesting that much of the model's ability to complete the task above the 50% random baseline is mediated by this single direction.

# 4 The Summarization Motif for Sentiment

Though we do not focus on circuit[4] analysis here, we note that initial patching experiments in the style of (Wang et al., 2022) revealed patterns which motivated our definition of the "summarization motif": when there is information (e.g. sentiment) stored at certain 'placeholder' or 'summary' tokens (e.g. commas, periods and certain nouns) despite these tokens not inherently having the information. Moreover, this information is causally significant for the model to complete a certain task (e.g. sentiment classification). We provide a detailed circuit-based analysis of this phenomenon in Appendix A.8. In this section, we focus on verifying this behaviour in Pythia-2.8b, and we replicate for other models in the Appendix (Table 5).

At first, we verify this phenomenon on toy datasets where we are able to isolate the effect using activation patching experiments. We find that in many cases this summarization results in a partial information bottleneck, in which the summarization points become as important (or sometimes more important) than the phrases containing the relevant information for sentiment tasks. Next, we reproduce these findings on natural text using the SST dataset (Section 2.1). We performed ablation experiments (Section 2.2) on comma positions. If comma representations do not summarize sentiment information, then our experiments should not damage the model's abilities. However, our results reveal a clear summarization motif for SST.

---

[3]We did this to maximise the chances of sentiment tokens occurring at similar positions

[4]We use the term "circuit" as defined by Wang et al. (2022), in the sense of a computational subgraph that is responsible for a significant proportion of the behavior of a neural network on some predefined task.

| | |
|---|---|
| **Original prompt** | Jack *loves* parties, ... Jack feels very |
| **Flipped prompt** | Jack *hates* parties, ... Jack feels very |
| **Freezing nodes** | Attention pattern, value vectors at commas |
| **Patching nodes** | Value vectors pre-comma, e.g. Jack loves parties |
| **Change in logit difference** | -38% |

(a) Isolating the effect of pre-comma phrases in ToyMoodStory

| | |
|---|---|
| **Original prompt** | Jack *loves* parties, ... Jack feels very |
| **Flipped prompt** | Jack *hates* parties, ... Jack feels very |
| **Freezing nodes** | Attention pattern |
| **Patching nodes** | Value vectors at commas and periods |
| **Change in logit difference** | -37% |

(b) Isolating the effect of commas in ToyMoodStory

| | |
|---|---|
| **Original prompt** | Jack *loves* parties ... Jack feels very |
| **Flipped prompt** | Jack *hates* parties, ... Jack feels very |
| **Freezing nodes** | Attention pattern |
| **Patching nodes** | All value vectors |
| **Change in logit difference** | -75% |

(c) Accumulating effects of commas and phrases in ToyMoodStory

Table 2: Patching experiments in ToyMoodStory, Pythia 2.8b. The similar results for 2a and 2b indicate that summarization information is comparably important as the original semantic information.

| | |
|---|---|
| **Original prompt** | Jack *loves* parties. [irrelevant text...] Jack feels very |
| **Flipped prompt** | Jack *hates* parties. [irrelevant text...] Jack feels very |
| **Freezing nodes** | Attention pattern, value vectors at periods |
| **Patching nodes** | Value vectors pre-period, e.g. Jack loves parties |

(a) Isolating the effect of pre-period phrases in ToyMoodStory

| | |
|---|---|
| **Original prompt** | Jack *loves* parties. [irrelevant text...] Jack feels very |
| **Flipped prompt** | Jack *hates* parties. [irrelevant text...] Jack feels very |
| **Freezing nodes** | Attention pattern |
| **Patching nodes** | Value vectors at periods |

(b) Isolating the effect of periods in ToyMoodStory

| Count of irrelevant tokens after preference phrase | Ratio of LD change for periods vs. phrases |
|---|---|
| 0 tokens | 0.29 |
| 10 tokens | 0.63 |
| 18 tokens | 0.92 |
| 22 tokens | 1.15 |

(c) Ratio between logit difference change for periods (3b) vs. pre-period (3a) phrases after patching values

Table 3: Patching experiments in ToyMoodStory with irrelevant text injection

**Summarization information is comparably important as original semantic information** In order to determine the extent of the information bottleneck presented by commas in sentiment processing, we tested the model's performance on ToyMoodStory (Section 2.1). We performed an activation patching experiment (Section 2.2) where we patched the attention value vectors at certain groups of token positions to flip the sentiment, along with the modification that we froze the model's attention patterns to ensure the model used the information from the patched commas in exactly the same way as it would have used the original information. Without this step, the model could simply avoid attending to the commas. Concretely, the three different interventions were:

1. Patching the value vectors at the pre-comma phrases (e.g., patching "John hates parties," with "John loves parties,") while freezing the value vectors at the commas and periods so they retain their original, unflipped values. This experiment (Table 2a) was designed to isolate the effect of the phrases, removing any reliance on punctuation tokens.

2. Patching the value vectors at the two commas and two periods alone. This experiment (Table 2b) was designed to isolate the effect of

the "summarization motif".

3. Patching *all* of the value vectors. This experiment (Table 2c) was designed to determine how the effects of the pre-comma phrases and commas accumulate to create the total effect of flipping the full phrase.

The experimental results (Table 2) show a similar drop in the logit difference for both the pre-comma and comma patching, demonstrating that fully half the effect of these phrases on the final logits for the correct tokens are mediated through the "summarization" motif. We continue to focus on results from Pythia-2.8b, but also replicated these findings across several models (Appendix, Table 5).

**Impact of summarization increases with distance** We also observed that reliance on summarization tends to increase with greater distances between the preference phrases and the final part of the prompt that would reference them. To test this, we injected irrelevant text[5] of varying sizes after each of the preference phrases in ToyMoodStory

---

[5]E.g. "John loves parties. *He has a red hat and wears it everywhere, especially when he is riding his bicycle through the city streets.* Mark hates parties. *He has a purple hat but only wears it on Sundays, when he takes his weekly walk around the lake.* One day, they were invited to a grand gala. John feels very"

| (a) | |
|---|---|
| Directions | DAS sentiment direction |
| Positions | All |
| Layers | All |
| Ablation type | Mean-ablation |
| Change in logit difference | −71% |
| Change in accuracy | −38% |

(a) Baselining the importance of the sentiment direction in SST

| (b) | |
|---|---|
| Directions | Random direction |
| Positions | All |
| Layers | All |
| Ablation type | Mean-ablation |
| Change in logit difference | < 1% |
| Change in accuracy | < 1% |

(b) Baselining the importance of random directions in SST

| (c) | |
|---|---|
| Directions | DAS sentiment direction |
| Positions | Commas |
| Layers | All |
| Ablation type | Mean-ablation |
| Change in logit difference | −18% |
| Change in accuracy | −18% |

(c) Isolating the sentiment axis information at commas in SST

| (d) | |
|---|---|
| Directions | Full Space |
| Positions | Commas |
| Layers | All |
| Ablation type | Mean-ablation |
| Change in logit difference | −17% |
| Change in accuracy | −19% |

(d) Isolating the importance of the full residual stream at commas in SST

Table 4: Ablation experiments in Stanford Sentiment Treebank (Section 2.1)

texts (after "John loves parties." etc.). We then computed a similar pair of logit difference metrics as depicted in 2, comparing the effect of patching value vectors at either the periods (3b) or the pre-period phrases (3a). Next we measured the ratio between these two logit difference changes for the periods vs. pre-period phrases, with higher values indicating more reliance on period summaries (3c). We found that the periods can be up to 15% **more** important than the actual phrases as this distance grows. Although these results are only a first step in assessing the importance of summarization relative to prompt length, they suggest this motif may become more significant as models grow in context length, and thus merits further study.

## 4.1 Summarization behavior in real-world datasets

**Data preparation** We appended the suffix "Review Sentiment:" to each of the prompts from SST and evaluated Pythia-2.8b on zero-shot classification according to whether positive or negative have higher probability, filtering to ensure these completions are in the top 10 tokens predicted. We then take the subset of examples that Pythia-2.8b classifies correctly that have at least one comma, which means we start with a baseline of 100% accuracy.

**Ablation baselines** We performed two baseline experiments in order to obtain a control for our later experiments. First to measure the total effect of the sentiment directions, we performed directional ablation (as described in 2.2) using the sentiment directions found with DAS, ablating along a single axis of the residual stream at every token position in every layer (4a), resulting in a 71% reduction in the logit difference and a 38% drop in accuracy (to 62% , where 50% is random chance). We also performed directional ablation on all tokens with a small set of random directions (4b), resulting in a < 1% change to the same metrics.

**Directional ablation at all comma positions** We then performed directional ablation–using the DAS sentiment direction (2.4) – to every comma in each prompt (4c), regardless of position, resulting in an 18% drop in the logit difference and an 18% drop in zero-shot classification accuracy. Comparing the latter result to the baseline from 4a indicates that nearly 50% of the model's sentiment-direction-mediated ability to perform the task accurately was mediated via sentiment information at the commas. We find this particularly significant because we did not take any special effort to ensure that commas were placed at the end of sentiment phrases.

**Mean-ablation of the full residual stream at all comma positions** Instead of relying on the sentiment direction computed using DAS as above, we also performed mean ablation experiments (2.2) on the full residual stream at comma positions. Specifically, we replaced each comma residual stream vector with the mean comma residual stream from the entire dataset in a layerwise fashion (4d). This resulted in a 17% drop in logit difference and an accuracy drop of 19% .

## 5 Conclusion

The two central novel findings of this research are the existence of a linear representation of sentiment and the use of summarization to store sentiment information. We have seen that the sentiment direction is causal and central to the circuitry of sentiment processing. Remarkably, this direction is so stark in the residual stream space that it can be found even with the most basic methods and on a tiny toy dataset, yet generalize to diverse real-world datasets. Summarization is a motif present in larger models with longer context lengths and greater proficiency in zero-shot classification. These summaries present a tantalising glimpse into the world-modelling behavior of transformers.

## Author Contributions

Oskar and Curt made equal contributions to this paper. Curt's focus was on circuit analysis and he discovered the summarization motif, leading to Section 4. Oskar was focused on investigating the direction and eventually conducted enough independent experiments to convince us that the direction was causally meaningful, leading to Section 3. Neel was our mentor as part of SERI MATS, suggested the initial project brief, and provided considerable mentorship during the research. Atticus acted an additional source of mentorship and guidance. His advice was particularly useful as someone with more of a background in causal mediation analysis than mechanistic interpretability.

### Acknowledgments
SERI MATS provided funding, lodging and office space for 2 months in Berkeley, California.

## Reproducibility Statement

To facilitate reproducibility of the results presented in this paper, we have provided detailed descriptions of the datasets, models, training procedures, algorithms, and analysis techniques used. We use publicly available models including GPT-2 and Pythia, with details on the specific sizes provided in Section 2.1. The methods for finding sentiment directions are described in full in Section 2.4. Our causal analysis techniques of activation patching, ablation, and directional patching are presented in Section 2.2. Circuit analysis details are extensively covered for two examples in Appendix A.8. The code for data generation, model training, and analyses has been prepared and documented and will be linked in the camera-ready version of this paper.

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

# A Appendix

## A.1 Related Work

**Sentiment Analysis** Understanding the emotional valence in text data is one of the first NLP tasks to be revolutionized by deep learning (Socher et al., 2013) and remains a popular task for benchmarking NLP models (Rosenthal et al., 2017; Nakov et al., 2016; Potts et al., 2021; Abraham et al., 2022). For a review of the literature, see (Pang and Lee, 2008; Liu, 2012; Grimes, 2014).

**Understanding Internal Representations** This research was inspired by the field of Mechanistic Interpretability, an agenda which aims to reverse-engineer the learned algorithms inside models (Olah et al., 2020; Elhage et al., 2021b; Nanda et al., 2023a). Exploring representations (Section 3) and world-modelling behavior inside transformers has garnered significant recent interest. This was studied in the context of synthetic game-playing models by Li et al. (2023) and evidence of linearity was demonstrated by Nanda (2023a) in the same context. Other work studying examples of world-modelling inside neural networks includes Li et al. (2021); Patel and Pavlick (2022); Abdou et al. (2021). Another framing of a very similar line of inquiry is the search for latent knowledge (Christiano et al., 2021; Burns et al., 2022). Prior to the transformer, representations of sentiment specifically were studied by Radford et al. (2017), notably, their finding of a sentiment neuron also implies a linear representation of sentiment.

**Causal Analysis of Language Models** We approach our experiments from a causal mediation analysis perspective. Our approach to identifying computational subgraphs that utilize feature representations as inspired by the 'circuits analysis' framework (Stefan Heimersheim, 2023; Varma et al., 2023; Hanna et al., 2023), especially the tools of mean ablation and activation patching (Vig et al., 2020; Geiger et al., 2021, 2023a; Meng et al., 2023; Wu et al., 2022, 2023; Wang et al., 2022; Conmy et al., 2023; Chan et al., 2023; Cohen et al., 2023). We use Distributed Alignment Search (Geiger et al., 2023b) in order to apply these ideas to specific subspaces.

## A.2 Limitations

Many of our casual abstractions do not explain 100% of sentiment task performance. There is likely circuitry we've missed, possibly as a result of distributed representations or superposition (Elhage et al., 2022) across components and layers. This may also be a result of self-repair behavior (Wang et al., 2022; McGrath et al., 2023). Patching experiments conducted on more diverse sentence structures could help to better isolate sentiment circuitry from more task-specific machinery.

The use of small datasets versus many hyperparameters and metrics poses a constant risk of gaming our own measures. Our results on the larger and more diverse SST dataset, and the consistent results across a range of models help us to be more confident in our conclusions.

Distributed Alignment Search (DAS) outperformed on most of our metrics but presents possible dangers of overfitting to a particular dataset and taking the activations out of distribution (Lange et al., 2023). We include simpler tools such as Logistic Regression as a sanity check on our findings. Ideally, we would love to see a set of best practices to avoid such illusions.

## A.3 Implications and future work

The summarization motif emerged naturally during our investigation of sentiment, but we would be very interested to study it in a broader range of contexts and understand what other factors of a particular model or task may influence the use of summarization.

When studying the circuitry of sentiment, we focused almost exclusively on attention heads rather than MLPs. However, early results suggest that further investigation of the role of MLPs and individual neurons is likely to yield interesting results (A.10).

## A.4 Impact Statement

This paper aims to advance the field of Mechanistic Interpretability. We see the long-term goal of this line of research as being able to help detect dangerous computation in language models such as *deception*. Even if the existence of a single "deception direction" in activation space seems a bit naive to postulate,

|  | DAS | KM | LR | MD | PCA | Random |
|---|---|---|---|---|---|---|
| **DAS** | 100.0% | 82.4% | 86.1% | 86.1% | 69.7% | 0.2% |
| **KM** | 82.4% | 100.0% | 95.1% | 95.7% | 80.0% | 1.7% |
| **LR** | 86.1% | 95.1% | 100.0% | 99.9% | 78.0% | 0.6% |
| **MD** | 86.1% | 95.7% | 99.9% | 100.0% | 79.5% | 0.6% |
| **PCA** | 69.7% | 80.0% | 78.0% | 79.5% | 100.0% | 0.7% |
| **Random** | 0.2% | 1.7% | 0.6% | 0.6% | 0.7% | 100.0% |

Figure A.1: Cosine similarity of directions learned by different methods in Pythia-2.8b's first layer. Each sentiment direction was derived from *adjective* representations in the ToyMovieReview dataset (Section 2.1).

| direction | accuracy |
|---|---|
| $k$-means | 86.4% |
| PCA | 82.2% |
| Mean Diff | 85.0% |
| LR | 90.5% |
| DAS | 80.8% |

Figure A.2: Accuracy using sentiment activations from the first residual stream layer of Pythia 2.8B to classify tokens as positive or negative. The threshold taken is the top/bottom 0.1% of activations over OpenWebText. Classification was performed by GPT-4.

hopefully in the future many of the tools developed here will help to detect representations of deception or of knowledge that the model is concealing, helping to prevent possible harms from LLMs.

### A.5 Further methods for finding directions

Using the same notation as in section 2.4, here are two further methods for computing a 'sentiment direction'.

**Mean Difference (MD)**  The direction is computed as $\frac{1}{|\mathbb{P}|} \sum_{p\in\mathbb{P}} a_p^L - \frac{1}{|\mathbb{N}|} \sum_{n\in\mathbb{N}} a_n^L$.

**Principal Component Analysis (PCA)**  The direction is the first component of $\{a_x^L : x \in \mathbb{P} \cup \mathbb{N}\}$.

**Convergence of five direction-finding methods**  We find high cosine similarity (Figure A.1) between the 5 different direction-finding methods. Note that cosine similarity is a potentially misleading metric in cases where the vectors can share a bias, but this is not a concern for a linear probe direction where there is no meaningful notion of a shared bias.

### A.6 Further evidence for a linear sentiment representation

### A.6.1 Clustering

In Section 2.4, we outline just a few of the many possible techniques for determining a direction which hopefully corresponds to sentiment. Is it overly optimistic to presume the existence of such a direction? The most basic requirement for such a direction to exist is that the residual stream space is clustered. We confirm this in two different ways.

(a) PCA on adjectives in and out of sample

(b) PCA on in-sample adjectives and out-of-sample verbs

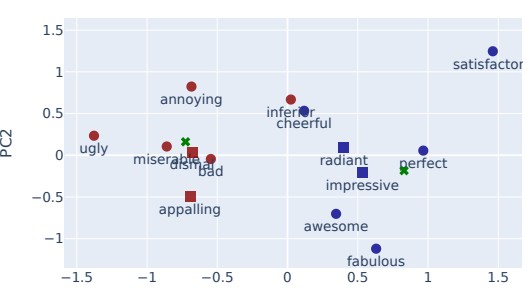
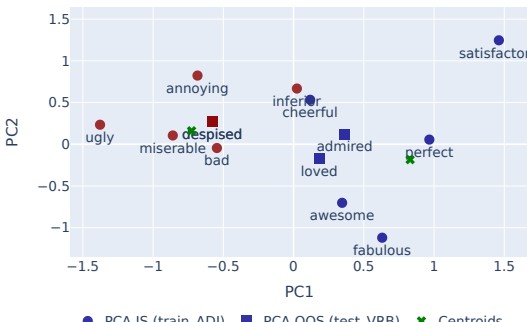

Figure A.3: 2-D PCA visualization of the embedding for a handful of adjectives and verbs (GPT2-small)

First we fit 2-D PCA to the token embeddings for a set of 30 positive and 30 negative adjectives. In Figure A.3, we see that the positive adjectives (blue dots) are very well clustered compared to the negative adjectives (red dots). Moreover, we see that sentiment words which are out-of-sample with respect to the PCA (squares) also fit naturally into their appropriate color. This applies not just for unseen adjectives (Figure A.3a) but also for verbs, an entirely out-of-distribution class of word (Figure A.3b).

Secondly, we evaluate the accuracy of 2-means trained on the Simple Movie Review Continuation adjectives (Section 2.1). The fact that we can classify in-sample is not very strong evidence, but we verify that we can also classify out-of-sample with respect to the $k$-means fitting process. Indeed, even on hold-out adjectives and on the verb tokens (which are totally out of distribution), we find that the accuracy is generally very strong across models. We also evaluate on a fully out of distribution toy dataset ("simple adverbs") of the form "The traveller [adverb] walked to their destination. The traveller felt very". The results can be found in Figure A.4. This is strongly suggestive that we are stumbling on a genuine representation of sentiment.

### A.6.2 Activation addition

We perform "activation addition" (Turner et al., 2023), i.e. we add a multiple of the sentiment direction to the first layer residual stream during each forward pass while generating sentence completions. We use GPT2-small for a single positive simple movie review continuation prompt: "I really enjoyed the movie, in fact I loved it. I thought the movie was just very...". We seek to verify that this can flip the generated outputs from positive to negative. The "steering coefficient" is the multiple of the sentiment direction which we add to the first layer residual stream.

By adding increasingly negative multiples of the sentiment direction, we find that indeed the completions become increasingly negative, without completely destroying the coherence of the model's generated text (Figure A.5). We are wary of taking the model's activations out of distribution using this technique, but we believe that the smoothness of the transition in combination with the knowledge of our findings in the patching setting give us some confidence that these results are meaningful.

### A.6.3 Multi-lingual sentiment

We use the first few paragraphs of Harry Potter in English and French as a standard text (Elhage et al., 2021b). We find that intermediate layers of Pythia-2.8b demonstrate intuitive sentiment activations for the French text (Figure A.6). It is important to note that none of the models are very good at French, but this was the smallest model where we saw hints of generalization to other languages. The representation was not evident in the first couple of layers, probably due to the poor tokenization of French words.

kmeans accuracy (gpt2-small)

| train_set | train_pos | train_layer | test_set | simple_test | simple_adverb |
| | | | test_pos | | |
| | | | ADJ | VRB | ADV |
|---|---|---|---|---|---|
| simple_train | ADJ | 0 | 100.0% | 83.3% | 50.0% |
| | | 1 | 100.0% | 100.0% | 55.3% |
| | | 2 | 100.0% | 100.0% | 60.5% |
| | | 3 | 100.0% | 100.0% | 65.8% |
| | | 4 | 100.0% | 100.0% | 78.9% |
| | | 5 | 100.0% | 100.0% | 57.9% |
| | | 6 | 100.0% | 100.0% | 84.2% |
| | | 7 | 100.0% | 100.0% | 71.1% |
| | | 8 | 100.0% | 100.0% | 65.8% |
| | | 9 | 100.0% | 100.0% | 68.4% |
| | | 10 | 91.7% | 100.0% | 60.5% |
| | | 11 | 91.7% | 100.0% | 60.5% |
| | | 12 | 33.3% | 58.3% | 31.6% |

(a) GPT-2 Small

kmeans accuracy (gpt2-medium)

| train_set | train_pos | train_layer | test_set | simple_test | simple_adverb |
| | | | test_pos | | |
| | | | ADJ | VRB | ADV |
|---|---|---|---|---|---|
| simple_train | ADJ | 0 | 100.0% | 100.0% | 50.0% |
| | | 1 | 100.0% | 83.3% | 50.0% |
| | | 2 | 100.0% | 100.0% | 47.4% |
| | | 3 | 91.7% | 100.0% | 47.4% |
| | | 4 | 91.7% | 100.0% | 47.4% |
| | | 5 | 100.0% | 100.0% | 47.4% |
| | | 6 | 100.0% | 100.0% | 68.4% |
| | | 7 | 91.7% | 100.0% | 50.0% |
| | | 8 | 91.7% | 100.0% | 84.2% |
| | | 9 | 100.0% | 100.0% | 86.8% |
| | | 10 | 100.0% | 100.0% | 71.1% |
| | | 11 | 100.0% | 100.0% | 94.7% |
| | | 12 | 100.0% | 100.0% | 65.8% |
| | | 13 | 100.0% | 100.0% | 63.2% |
| | | 14 | 100.0% | 100.0% | 73.7% |
| | | 15 | 100.0% | 100.0% | 60.5% |
| | | 16 | 100.0% | 100.0% | 57.9% |
| | | 17 | 100.0% | 100.0% | 55.3% |
| | | 18 | 100.0% | 100.0% | 55.3% |
| | | 19 | 100.0% | 100.0% | 76.3% |
| | | 20 | 100.0% | 100.0% | 84.2% |
| | | 21 | 100.0% | 91.7% | 65.8% |
| | | 22 | 100.0% | 100.0% | 52.6% |
| | | 23 | 100.0% | 100.0% | 57.9% |
| | | 24 | 83.3% | 58.3% | 50.0% |

(b) GPT-2 Medium

Figure A.4: 2-means classification accuracy for various GPT-2 sizes, split by layer (showing up to 24 layers)

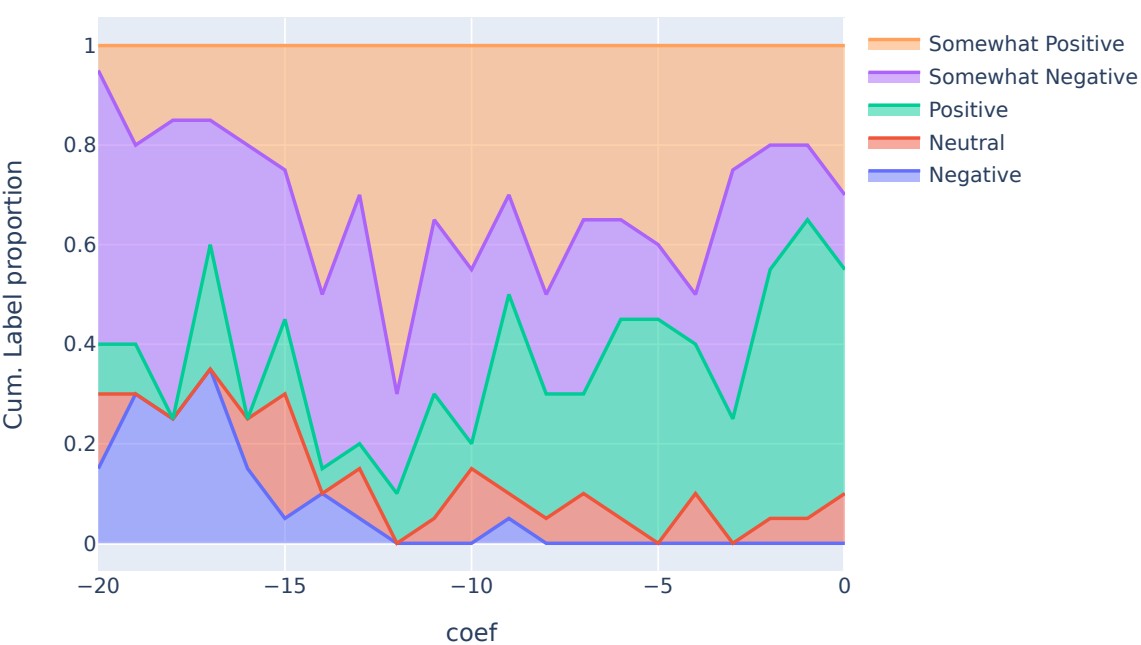

Figure A.5: Area plot of sentiment labels for generated outputs by activation steering coefficient, starting from a single positive movie review continuation prompt. Activation addition (Turner et al., 2023) was performed in GPT2-small's first residual stream layer. Classification was performed by GPT-4.

<|endoftext|>

Mr. and Mrs. Dursley, of number four, Privet Drive, were proud to say that they were perfectly normal, thank you very much. They were the last people you'd expect to be involved in anything strange or mysterious, because they just didn't hold with such nonsense.

Mr. Dursley was the director of a firm called Grunnings, which made drills. He was a big, beefy man with hardly any neck, although he did have a very large mustache. Mrs. Dursley was thin and blonde and had nearly twice the usual amount of neck, which came in very useful as she spent so much of her time craning over garden fences, spying on the neighbors. The Dursleys had a small son called Dudley and in their opinion there was no finer boy anywhere.

The Dursleys had everything they wanted, but they also had a secret, and their greatest fear was that somebody would discover it. They didn't think they could bear it if anyone found out about the Potters. Mrs. Potter was Mrs. Dursley's sister, but they hadn't met for several years; in fact, Mrs. Dursley pretended she didn't have a sister, because her sister and her good-for-nothing husband were as unDursleyish as it was possible to be. The Dursleys shuddered to think what the neighbors would say if the Potters arrived in the street. The Dursleys knew that the Potters had a small son, too, but they had never even seen him. This boy was another good reason for keeping the Potters away; they didn't want Dudley mixing with a child like that.

When Mr. and Mrs. Dursley woke up on the dull, gray Tuesday our story starts, there was nothing about the cloudy sky outside to suggest that strange and mysterious things would soon be happening all over the country. Mr. Dursley hummed as he picked out his most boring tie for work, and Mrs. Dursley gossiped away happily as she wrestled a screaming Dudley into his high chair.

(a) First 4 paragraphs of Harry Potter in English

<|endoftext|>

Mr et Mrs Dursley, qui habitaient au 4, Privet Drive, avaient toujours affirmé avec la plus grande fierté qu'ils étaient parfaitement normaux, merci pour eux. Jamais quiconque n'aurait imaginé qu'ils puissent se trouver impliqués dans quoi que ce soit d'étrange ou de mystérieux. Ils n'avaient pas de temps à perdre avec des sornettes.

Mr Dursley dirigeait la Grunnings, une entreprise qui fabriquait des perceuses. C'était un homme grand et massif, qui n'avait pratiquement pas de cou, mais possédait en revanche une moustache de belle taille. Mrs Dursley, quant à elle, était mince et blonde et disposait d'un cou deux fois plus long que la moyenne, ce qui lui était fort utile pour espionner ses voisins en regardant par-dessus les clôtures des jardins. Les Dursley avaient un petit garçon prénommé Dudley et c'était à leurs yeux le plus bel enfant du monde.

Les Dursley avaient tout ce qu'ils voulaient. La seule chose indésirable qu'ils possédaient, c'était un secret dont ils craignaient plus que tout qu'on le découvre un jour. Si jamais quiconque venait à entendre parler des Potter, ils étaient convaincus qu'ils ne s'en remettraient pas. Mrs Potter était la soeur de Mrs Dursley, mais toutes deux ne s'étaient plus revues depuis des années. En fait, Mrs Dursley faisait comme si elle était fille unique, car sa soeur et son bon à rien de mari étaient aussi éloignés que possible de tout ce qui faisait un Dursley. Les Dursley tremblaient d'épouvante à la pensée de ce que diraient les voisins si par malheur les Potter se montraient dans leur rue. Ils savaient que les Potter, eux aussi, avaient un petit garçon, mais ils ne l'avaient jamais vu. Son existence constituait une raison supplémentaire de tenir les Potter à distance: il n'était pas question que le petit Dudley se mette à fréquenter un enfant comme celui-là.

Lorsque Mr et Mrs Dursley s'éveillèrent, au matin du mardi où commence cette histoire, il faisait gris et triste et rien dans le ciel nuageux ne laissait prévoir que des choses étranges et mystérieuses allaient bientôt se produire dans tout le pays. Mr Dursley fredonnait un air en nouant sa cravate la plus sinistre pour aller travailler et Mrs Dursley racontait d'un ton badin les derniers potins du quartier en s'efforçant d'installer sur sa chaise de bébé le jeune Dudley qui braillait de toute la force de ses poumons.

(b) First 3 paragraphs of Harry Potter in French

Figure A.6: First paragraphs of Harry Potter in different languages. Model: Pythia-2.8b.

### A.6.4 Universality examples

For comparison with Figures A.1, 2 and Table 1, we include Figure A.7a, Figure A.8 and Figure A.7 where we visualise the similarity and classification accuracy of directions found by different methods, this time for GPT2-small (Section 2.1), StableLM 3B (Tow, 2023), Gemma 2B (Team et al., 2024) and Qwen 1.8B (Bai et al., 2023) instead of Pythia-2.8b.

### A.6.5 Generalization at intermediate layers

If the sentiment direction was simply a trivial feature of the token embedding, then one might expect that directional patching would be most effective in the first or final layer. However, we see in Figure A.9 that in fact it is in intermediate layers of the model where we see the strongest out-of-distribution performance on SST. This suggests the speculative hypothesis that the model uses the residual stream to form abstract concepts in intermediate layers and this is where the latent knowledge of sentiment is most prominent.

### A.7 Limitations to our linearity claim

Did we find a truly universal sentiment direction, or merely the first principal component of directions used across different sentiment tasks? As found by Bricken et al. (2023), we suspect that this feature could be "split" further into more specific sentiment features. We performed an experiment to help validate that the common sentiment feature across tasks is one dimensional. DAS can be used not just to find a causally impactful direction, but a causal subspace of any dimension. Figure A.10 demonstrates that whilst increasing the DAS dimension improves the patching metric in-sample (A.10a), the metric does not improve out-of-distribution (A.10b).

Similarly, one might wonder if there is really a single bipolar sentiment direction or if we have simply found the difference between a "positive" and a "negative" sentiment direction. It turns out that this distinction is not well-defined, given that we find empirically that there is a direction corresponding to "valenced words". Indeed, if $x$ is the valence direction and $y$ is the sentiment direction, then $p = x + y$ represents positive sentiment and $n = x - y$ is the negative direction. Conversely, we can reframe as starting from the positive/negative directions $p$ and $n$, and then re-derive $x = \frac{p+n}{2}$ and $y := \frac{p-n}{2}$.

### A.8 Detailed circuit analysis

In order to build a picture of each circuit, we used the process pioneered in Wang et al. (2022):

- Identify which model components have the greatest impact on the logit difference when path patching is applied (with the final result of the residual stream set as the receiver).

- Examine the attention patterns (value-weighted, in some cases) and other behaviors of these components (in practice, attention heads) in order to get a rough idea of what function they are performing.

- Perform path-patching using these heads (or a distinct cluster of them) as receivers.

- Repeat the process recursively, performing contextual analyses of each "level" of attention heads in order to understand what they are doing, and continuing to trace the circuit backwards.

In each path-patching experiment, change in logit difference is used as the patching metric. We started with GPT-2 as an example of a classic LLM displays a wide range of behaviors of interest, and moved to larger models when necessary for the task we wanted to study (choosing, in each case, the smallest model that could do the task).

### A.8.1 Simple sentiment - GPT-2 small

In this sub-section, we present an overview of circuit findings that give qualitative hints of the summarization motif, and restrict quantitative analysis of the summarization motif to 4.

We examined the circuit performing the ToyMovie review task, i.e. for the following sentence template: "I thought this movie was ADJECTIVE, I VERBed it. Conclusion: This movie is". Mechanistically, this is a binary classification task, and a naive hypothesis is that attention heads attend directly from the final token (which we label 'END') to the valenced tokens (the adjective token, ADJ, and the verb token VRB) and map

|  | DAS | K_means | LR | Mean_diff | PCA | Random |
|---|---|---|---|---|---|---|
| DAS | 100.0% | 72.6% | 87.1% | 86.9% | 79.9% | 2.4% |
| K_means | 72.6% | 100.0% | 79.4% | 83.1% | 88.0% | 1.7% |
| LR | 87.1% | 79.4% | 100.0% | 99.1% | 90.2% | 0.5% |
| Mean_diff | 86.9% | 83.1% | 99.1% | 100.0% | 94.6% | 0.5% |
| PCA | 79.9% | 88.0% | 90.2% | 94.6% | 100.0% | 1.2% |
| Random | 2.4% | 1.7% | 0.5% | 0.5% | 1.2% | 100.0% |

(a) GPT2-small

|  | DAS | K_means | LR | Mean_diff | Random |
|---|---|---|---|---|---|
| DAS | 100.0% | 55.0% | 51.1% | 63.0% | 2.4% |
| K_means | 55.0% | 100.0% | 45.8% | 84.9% | 0.9% |
| LR | 51.1% | 45.8% | 100.0% | 81.4% | 2.2% |
| Mean_diff | 63.0% | 84.9% | 81.4% | 100.0% | 0.7% |
| Random | 2.4% | 0.9% | 2.2% | 0.7% | 100.0% |

(b) Gemma-2B

|  | DAS | K_means | LR | Mean_diff | Random |
|---|---|---|---|---|---|
| DAS | 100.0% | 58.5% | 80.3% | 80.3% | 3.1% |
| K_means | 58.5% | 100.0% | 68.5% | 69.9% | 5.0% |
| LR | 80.3% | 68.5% | 100.0% | 100.0% | 4.4% |
| Mean_diff | 80.3% | 69.9% | 100.0% | 100.0% | 4.5% |
| Random | 3.1% | 5.0% | 4.4% | 4.5% | 100.0% |

(c) Qwen-1.8B

|  | DAS | K_means | LR | Mean_diff | PCA | Random |
|---|---|---|---|---|---|---|
| DAS | 100.0% | 29.5% | 68.8% | 68.7% | 54.6% | 1.3% |
| K_means | 29.5% | 100.0% | 47.6% | 49.5% | 79.7% | 2.0% |
| LR | 68.8% | 47.6% | 100.0% | 99.9% | 78.6% | 1.8% |
| Mean_diff | 68.7% | 49.5% | 99.9% | 100.0% | 80.8% | 1.8% |
| PCA | 54.6% | 79.7% | 78.6% | 80.8% | 100.0% | 1.6% |
| Random | 1.3% | 2.0% | 1.8% | 1.8% | 1.6% | 100.0% |

(d) StableLM-3B

Figure A.7: Cosine similarity of directions learned by different methods in the first layer residual stream of different models. Each sentiment direction was derived from *adjective* representations in the ToyMovieReview dataset (Section 2.1).

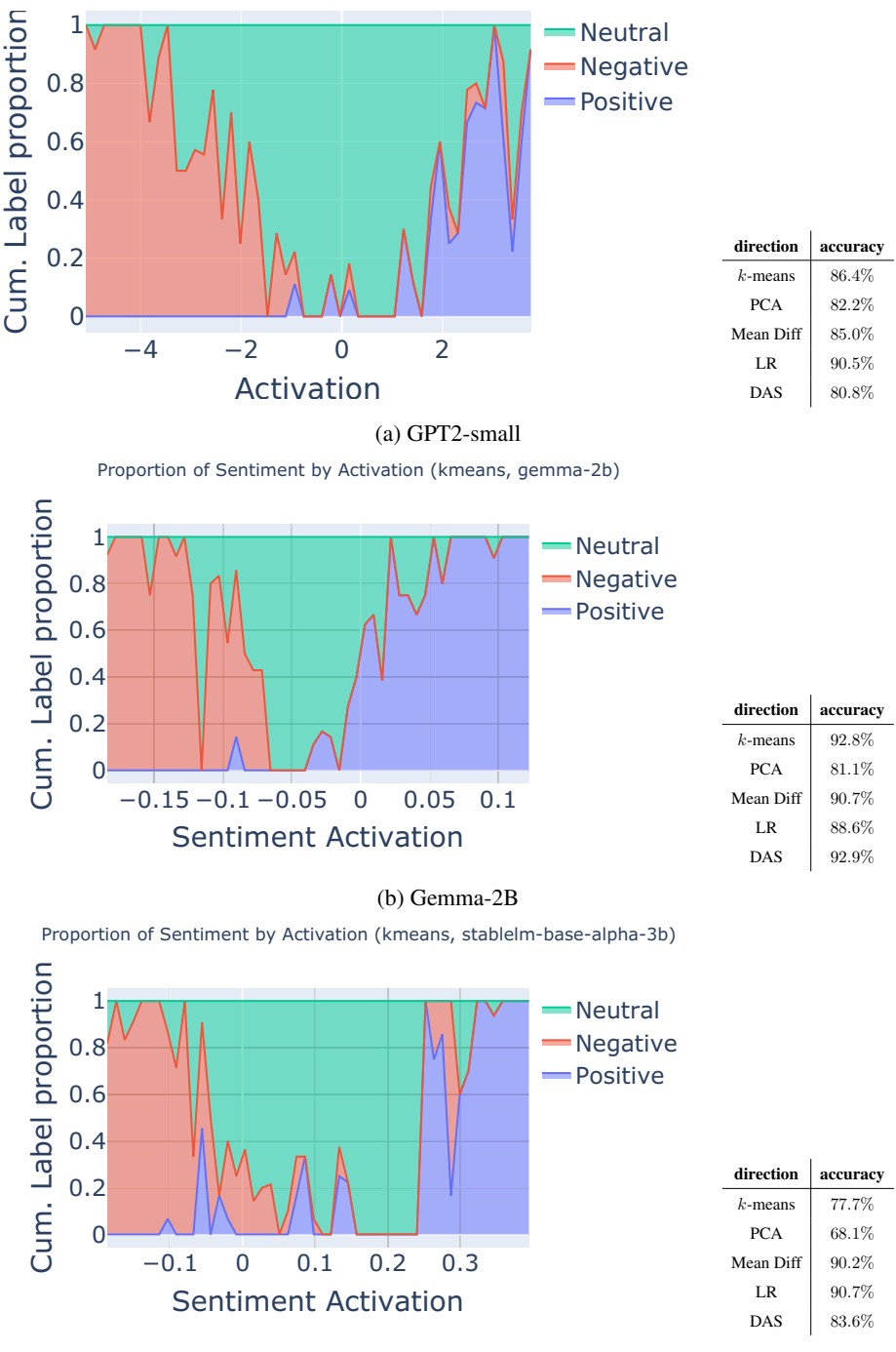

Figure A.8: Area plot of sentiment labels for OpenWebText samples by sentiment activation, i.e. the projection of the first residual stream layer at that token onto the sentiment direction (left). Accuracy using sentiment activations to classify tokens as positive or negative (right). The threshold taken is the top/bottom 0.1% of activations over OpenWebText. Classification was performed by GPT-4.

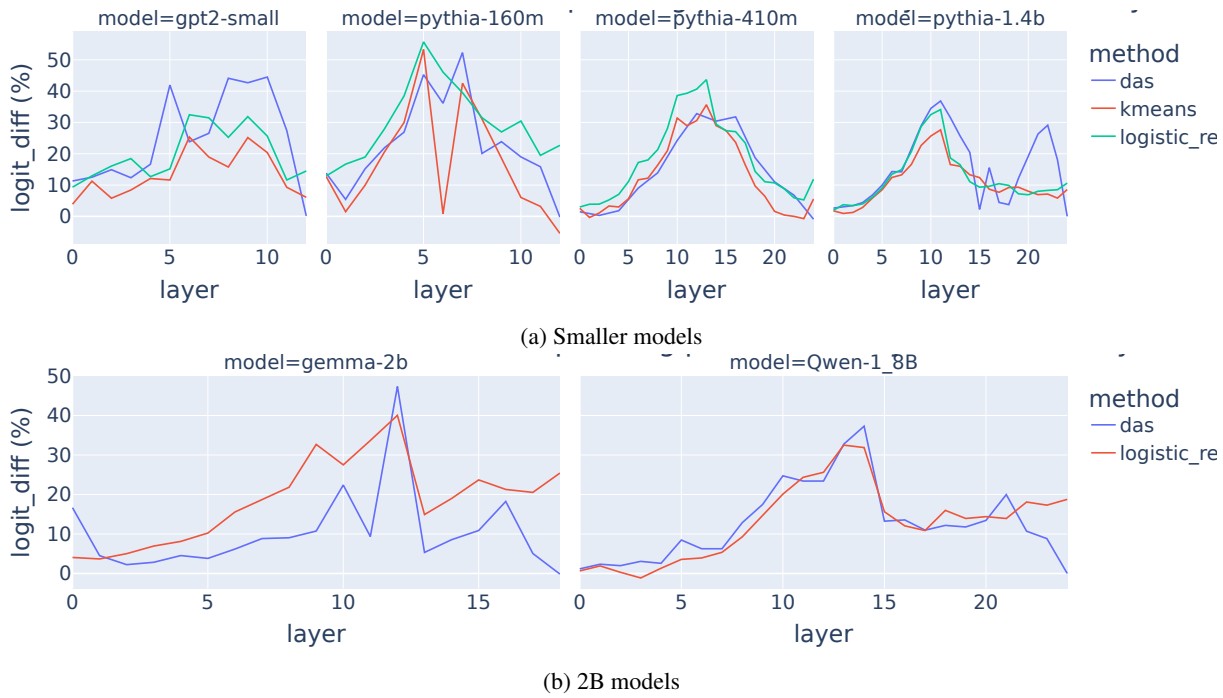

(a) Smaller models

(b) 2B models

Figure A.9: Patching results for directions trained on toy datasets and evaluated on the Stanford Sentiment Treebank test partition. We tend to find the best generalization when training and evaluating at a layer near the middle of the model. We scaffold the prompt using the suffix Overall the movie was very and compute the logit difference between good and bad. The patching metric (y-axis) is then the % mean change in logit difference.

positive sentiment to positive outputs and vice versa. This does happen but it is not the only mechanism. Attention head output is causally important at intermediate token positions (in particular, the final 'movie' token, SUM), which are then read from when producing output at END. We consider this an instance of summarization, in which the model aggregates causally-important information relating to an entity at a particular token for later usage, rather than simply attending back to the original tokens that were the source of the information.

Using a threshold of 5%-or-greater damage to the logit difference for our patching experiments, we found that GPT-2 Small contained 4 primary heads contributing to the most proximate level of circuit function–10.4, 9.2, 10.1, and 8.5 (using "layer.head" notation). Examining their value-weighted attention patterns, we found that attention to ADJ and VRB in the sentence was most prominent in the first three heads, but 8.5 attended primarily to the second "movie" token. We also observed that 9.2 attended to this token as well as to ADJ. We label 8.5 and 9.2 as "summary readers", and the second "movie" token as the SUM token (as in "summary"). (Results of activation patching can be seen in Fig. A.12.)

Conducting path-patching with 8.5 and 9.2 as receivers, we identified two heads–7.1 and 7.5–that primarily attend to ADJ and VRB from the "movie" token. We further determined that the output of these heads, when path-patched through 9.2 and 8.5 as receivers, was causally important to the circuit (with patching causing a logit difference shift of 7% and 4% respectively for 7.1 and 7.5). Hence we label 7.1 and 7.5 as "summary writers". This was not the case for other token positions, which demonstrates that causally relevant information is indeed being specially written to the SUM token, as suggested by our choice of label.

Repeating our analysis with lower thresholds yielded more heads with the same behavior but weaker effect sizes, adding 9.10, 11.9, and 6.4 as summary reader, direct sentiment reader, and sentiment summarizer respectively. This gives a total of 9 heads making up the circuit.

In summary, these results suggest that there is a circuit made up of 9 attention heads accomplishing the task as follows:

1. Identify sentiment-laden words in the prompt, at ADJ and VRB.

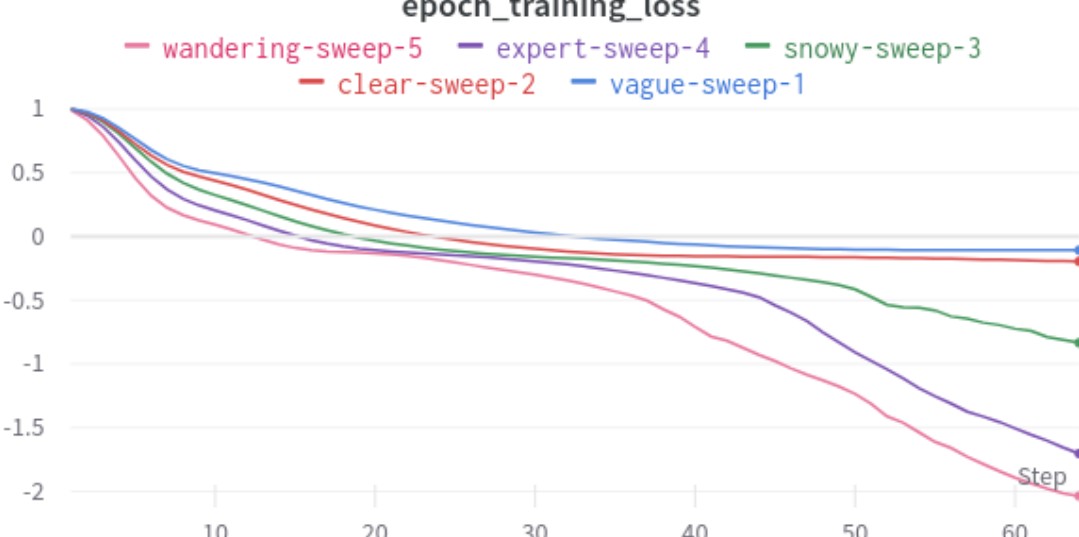

(a) Training loss for DAS on adjectives in a toy movie review dataset

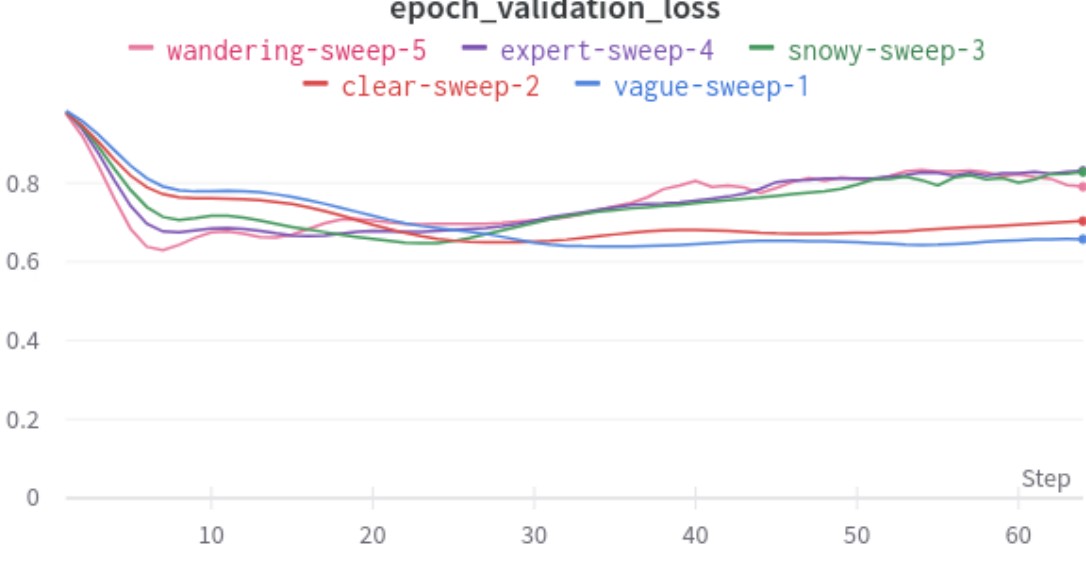

(b) Validation loss for DAS on a simple character mood dataset with a varying adverb

Figure A.10: DAS sweep over the subspace dimension (GPT2-small). The runs are labelled with the integer $n$ where $d_{\mathrm{DAS}} = 2^{n-1}$. Loss is 1 minus the usual patching metric.

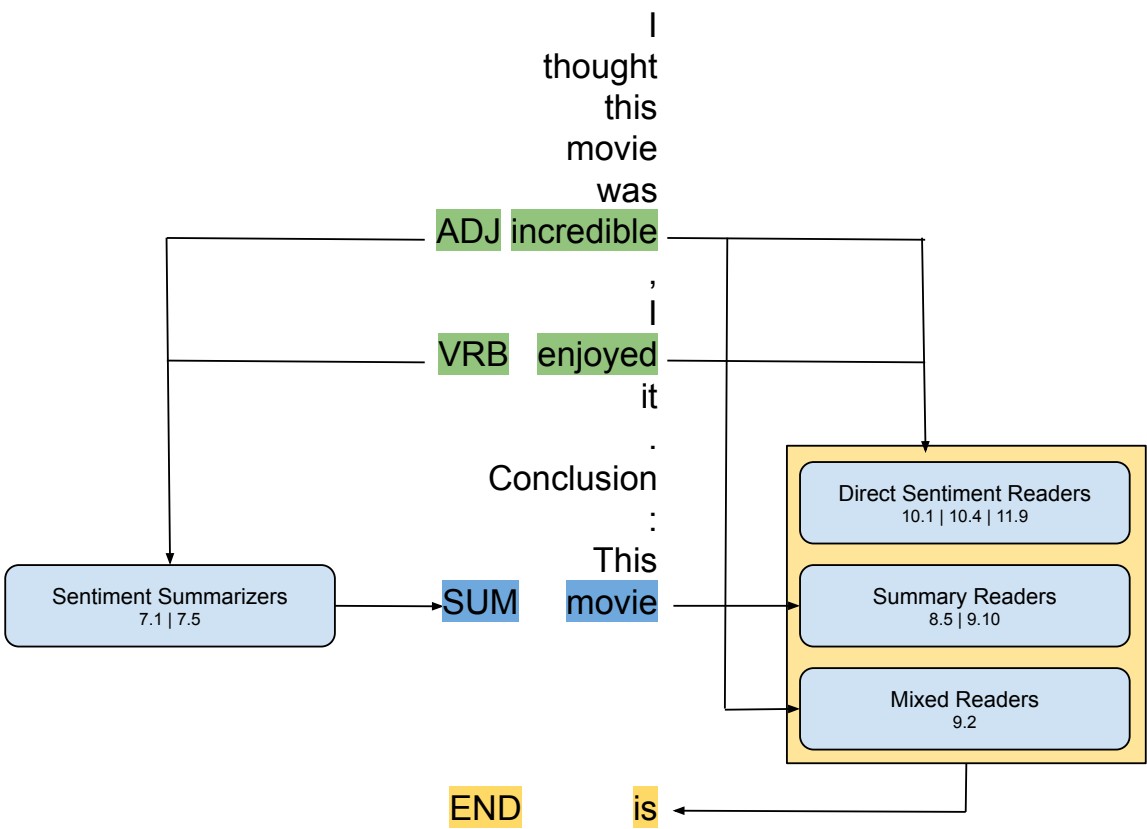

Figure A.11: Primary components of GPT-2 sentiment circuit for the ToyMovieReview dataset. Here we can see both direct use of sentiment-laden words in predicting sentiment at END as well as an example of the summarization motif at the SUM position (the final 'movie' token). Heads 7.1 and 7.5 write to this position and this information is causally relevant to the contribution of the summary readers at END.

2. "Summary writer" attention heads write out sentiment information to SUM (the final "movie" token).

3. "Summary reader" attention heads read from ADJ, VRB and SUM and write to END.[6]

To further validate this circuit and the involvement of the sentiment direction, we patched the entirety of the circuit at the ADJ and VRB positions along the sentiment direction only, achieving a 58.3% rate of logit flips and a logit difference drop of 54.8% (in terms of whether a positive or negative next token was predicted). Patching the circuit at those positions along all directions resulted in flipping 97% of logits and a logit difference drop of 75%, showing that the sentiment direction is responsible for the majority of the function of the circuit.

### A.8.2 ToyMoodStory circuit - Pythia-2.8b

We next examined the circuit that processes the ToyMoodStory dataset (Section 2.1) in Pythia-2.8b, the smallest model that could perform this more complex task that requires more summarization. The sentence template is Carl hates parties, and avoids them whenever possible. Jack loves parties, and joins them whenever possible. One day, they were invited to a grand gala. Jack feels very [excited/nervous]. We did not attempt to reverse-engineer the entire circuit, but examined it from the perspective of what matters causally for sentiment processing–especially determining to what extent summarization occurred.

---

[6]We note that our patching experiments indicate that there is no causal dependence on the output of other model components at the ADJ and VRB positions–only at the SUM position.

[7]That is, the attention pattern weighted by the norm of the value vector at each position as per Kobayashi et al. (2020). We favor this over the raw attention pattern as it filters for *significant* information being moved.

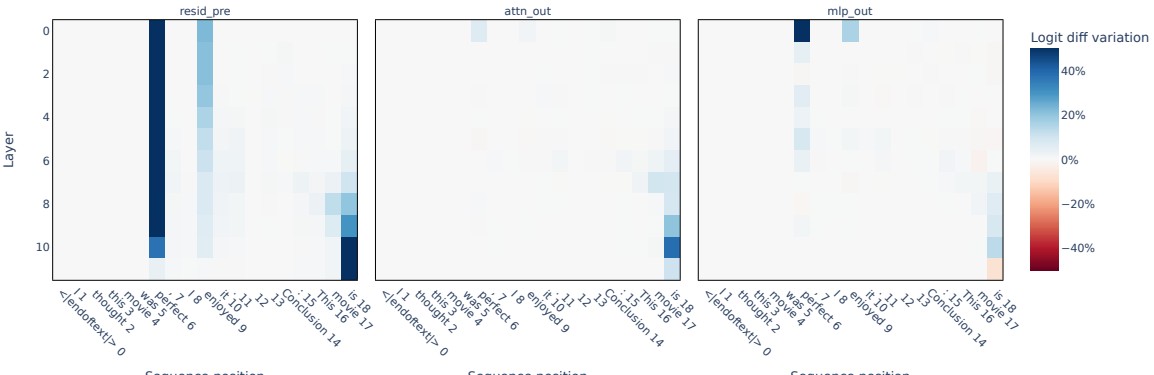

Figure A.12: Activation patching results for the GPT-2 Small ToyMovieReview circuit, showing how much of the original logit difference is recaptured when swapping in activations from $x_{orig}$ (when the model is otherwise run on $x_{flipped}$). Note that attention output is only important at the SUM position, and that this information is important to task performance at the residual stream layers (8 and 9) in which the summary-readers reside. Other than this, the most important residual stream information lies at the ADJ and VRB positions.

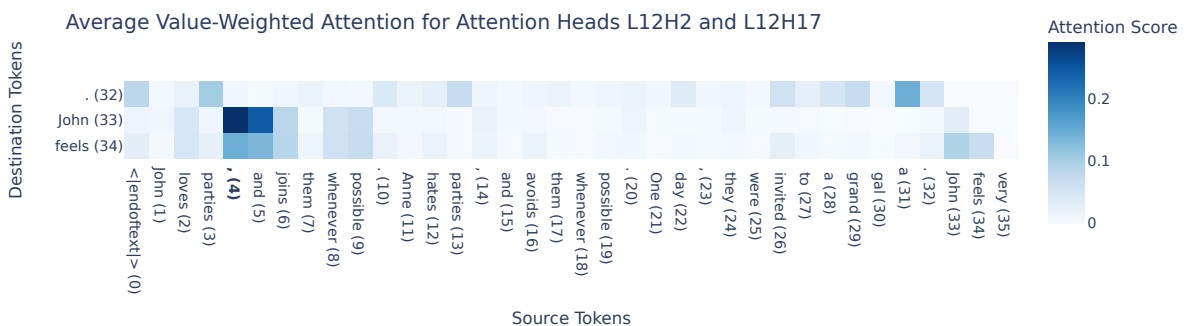

Figure A.13: Value-weighted[7] averaged attention to commas and comma phrases in Pythia-2.8b from the top two attention heads writing to the repeated name and "feels" tokens–two key components of the summarization sub-circuit in the ToyMoodStories task. Note that they attend heavily to the relevant comma from both destination positions.

Following the same process as with GPT-2 with preference/sentiment-flipped prompts (that is, taking $x_{orig}$ to be "John hates parties,... Mary loves parties," and $x_{flipped}$ to be "John loves parties,... Mary hates parties"), we initially identified 5 key heads that were most causally important to the logit difference at END: 17.19, 22.5, 14.4, 20.10, and 12.2 (in "layer.head" notation). Examining the value-weighted attention patterns, we observed that the top token receiving attention from END was always the repeated name RNAME (e.g., "John" in "John feels very") or the "feels" token FEEL, indicating that some summarization may have taken place there.

We also observed that the top token attended to from RNAME and FEEL was in fact the comma at the end of the queried preference phrase (that is, the comma at the end of "John hates parties"). We designate this position COMMASUM.

**Multi-functional heads** Interestingly, we observed that most of these heads were multi-functional: that is, they both attended to COMMASUM from RNAME and FEEL, and also attended to RNAME and FEEL from END, producing output in the direction of the logit difference. This is possible because these heads exist at different layers, and later heads can read the summarized information from previous heads as well as writing their own summary information.

**Direct effect heads** Specifically, the direct effect heads were:

- Head 17.19 did not attend to commas significantly, but did attend to the periods at the end of each preference sentence in addition to its primary attention to RNAME and FEEL, and did not display COMMASUM-reading behavior.

- Head 22.5 attended almost exclusively to FEEL, and did not display COMMASUM-reading behavior.

- Other direct effect heads (14.4, 20.10 and 12.2) did show COMMASUM-reading behavior as well as reading from the near-end tokens to produce output in the direction of the logit difference. In each case, we verified with path-patching that information from these positions was causally relevant.

**Name summary writers**   We also found important heads (12.17 being by far the most important) that are only engaged with attending to COMMASUM and producing output at RNAME and FEEL.

**Comma summary writers**   We further investigated what circuitry was causally important to task performance mediated through the COMMASUM positions, but did not flesh this out in full detail; after finding initial examples of summarization, we focused on its causal relevance and interaction with the sentiment direction, leaving deeper investigation to future work.

**Overview of heads**   In summary, the three main attention heads involved in this circuit were as follows.

- "Comma-reading heads": A set of attention heads **attended primarily to the comma** following the preference phrase for the queried subject (e.g. John hates parties,), and secondarily to other words in the phrase, as seen in Figure A.13. We observed this phenomenon both with regular attention and value-weighted attention, and found via path patching that **these heads relied primarily on the comma token** for their function, as seen in Figure A.15.

- "Name-writing heads": Heads attending to preference phrases (e.g., the entirety of "John loves parties," including the final comma) tended to write to the repeated name token near the end of the sentence (John) as well as to the feels token–another type of summarization behavior.

- "Name-reading heads": Later heads attended to the repeated name and feels tokens, affecting the output logits at END.

### A.9   Additional summarization findings

#### A.9.1   Circuitry for processing commas vs. original phrases is semi-separate

Though there is overlap between the attention heads involved in the circuitry for processing sentiment from key phrases and that from summarization points, there are also some clear differences, suggesting that the ability to read summaries could be a specific capability developed by the model (rather than the model simply attending to high-sentiment tokens).

As can be seen in Figure A.14, there are distinct groups of attention heads that result in damage to the logit difference in different situations–that is, some react when phrases are patched, some react disproportionately to comma patching, and one head seems to have a strong response for either patching case. This is suggestive of semi-separate summary-reading circuitry, and we hope future work will result in further insights in this direction.

#### A.9.2   Results from other models

We replicated the ToyMoodStories comma-swapping experiment (as explained in Section 4) in Pythia-6.9b and Mistral-7b as well as two Gemma and two Qwen models, with results shown in Table 5.

| Intervention | Pythia-2.8b | Pythia-6.9b | Mistral-7b | Gemma-2b | Gemma-7b | Qwen-1.8b | Qwen-7b |
|---|---|---|---|---|---|---|---|
| Patching full phrase values (incl. commas) | -75% | -152% | -155% | -152% | -120% | -181% | -145% |
| Patching pre-comma values (freezing commas & periods) | -38% | -46% | -16% | -68% | -42% | -71% | -32% |
| Patching comma and period values only | -37% | -68% | -100% | -42% | -52% | -72% | -36% |

Table 5: Change in logit difference from patching at commas in ToyMoodStory in three different models

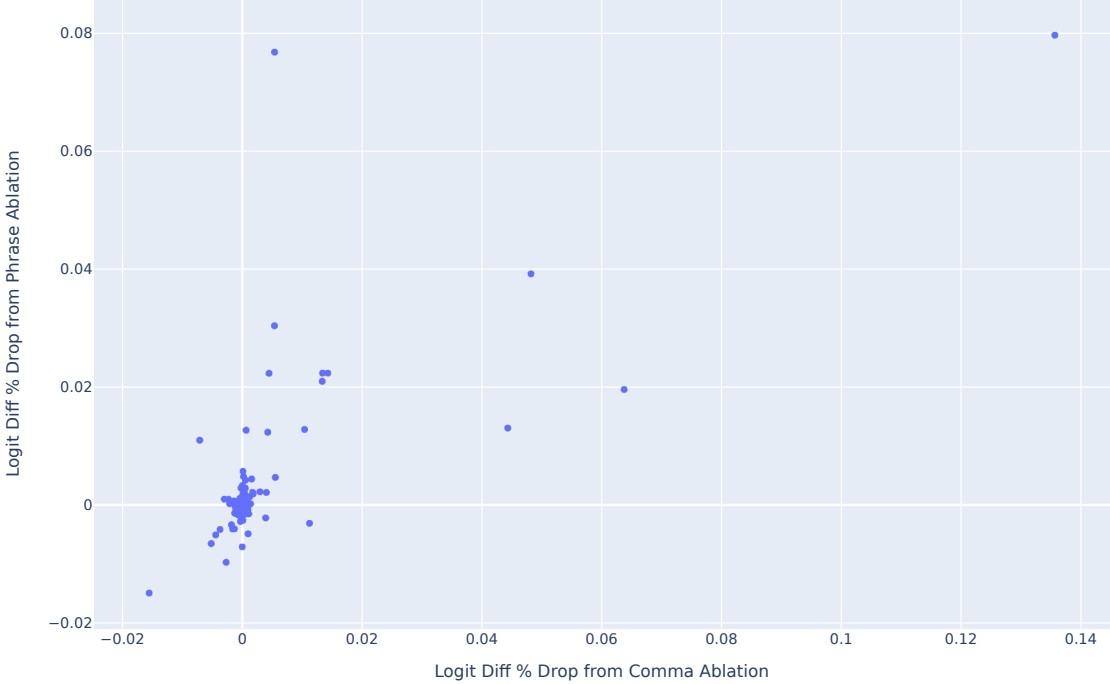

Figure A.14: Logit difference drops by head when commas or pre-comma phrases are patched. Model: Pythia-2.8b.

We take this as evidence that the comma-summarization phenomenon is not limited exclusively to Pythia-2.8b.

### A.10 Neurons writing to sentiment direction in GPT2-small are interpretable

We observed that the cosine similarities of neuron out-directions with the sentiment direction are extremely heavy tailed (Figure A.16). Thanks to Neuroscope (Nanda, 2023b), we can quickly see whether these neurons are interpretable. Indeed, here are a few examples from the tails of that distribution:

- L3N1605 activates on "hesitate" following a negation

- Neuron L6N828 seems to be activating on words like "however" or "on the other hand" *if* they follow something negative

- Neuron L5N671 activates on negative words that follow a "not" contraction (e.g. didn't, doesn't)

- L6N1237 activates strongly on "but" following "not bad"

We take L3N1605, the "not hesitate" neuron, as an extended example and trace backwards through the network using Direct Logit Attribution[8]. We computed the relative effect of different model components on L3N1605 in the two different cases "I would not hesitate" vs. "I would always hesitate". The main contributors to this difference are L1H0, L3H10, L3H11 and MLP2. Expanding out MLP2 into individual neurons we find that the contributions to L3N1605 are sparse. For example, L2N1154 activates on words like "don't", "not", "no", etc. It activates on "not" but not "hesitate" in "I would not hesitate" but activates on "hesitate" in "I would always hesitate". Visualizing the attention pattern of L1H0 shows that it attends from "hesitate" to the previous token if it is "not", but not if it is "always".

---

[8]This technique decomposes model outputs into the sum of contributions of each component, using the insight from Elhage et al. (2021b) that components are independent and additive

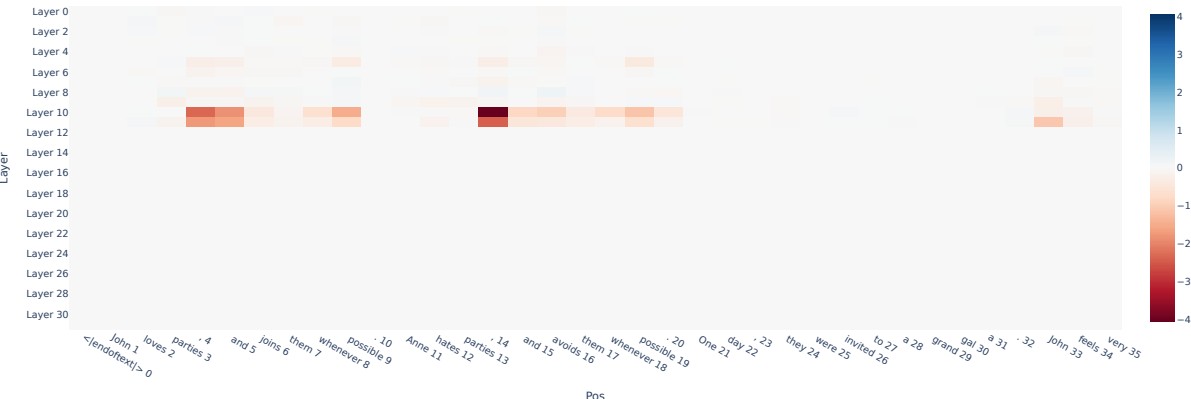

Figure A.15: Path-patching commas and comma phrases in Pythia-2.8b, with attention heads L12H2 and L12H17 writing to repeated name and "feels" as receivers. Patching the paths between the comma positions and the receiver heads results in the greatest performance drop for these heads.

These anecdotal examples suggest at a complex network of machinery for transmitting sentiment information across components of the network using a single critical axis of the residual stream as a communication channel. We think that exploring these neurons further could be a very interesting avenue of future research, particularly for understanding how the model updates sentiment based on negations where these neurons seem to play a critical role.

## A.11   Glossary

**Glossary**

**ablation**  A technique where we eliminate the contribution of a particular component to a model's output (usually by replacing the component's output with zeros or the mean over some dataset or a random sample from some dataset) in order to demonstrate the magnitude of its importance. (See Section 2.2)

**activation addition**  Formerly called "activation steering", a technique from Turner et al. (2023) where a vector is added to the residual stream at a certain position (or all positions) and layer during each forward pass while generating sentence completions. In our case, the vector is the sentiment direction.

**activation patching**  A technique introduced in Meng et al. (2023), under the name 'causal tracing', which uses an intervention to identify which activations in a model matter for producing some output. It runs the model on some 'clean' input, replaces (patches) an activation with that same activation on 'flipped' input, and sees how much that shifts the output from 'clean' to 'flipped'. (See Section 2.2)

**activation steering**  See activation addition.

**circuit**  A computational subgraph of a neural network which performs some human-interpretable task (Wang et al., 2022).

**DAS**  Distributed Alignment Search (Geiger et al., 2023b) uses gradient descent to train a rotation matrix representing an orthonormal change of basis to one better aligned with the model's features. We mostly focus on a special case of finding a singular critical direction, where we patch along the first dimension of the rotated basis and then use a smooth patching metric (such as the logit difference between positive and negative completions) as the objective to be minimised. (See Section 2.4)

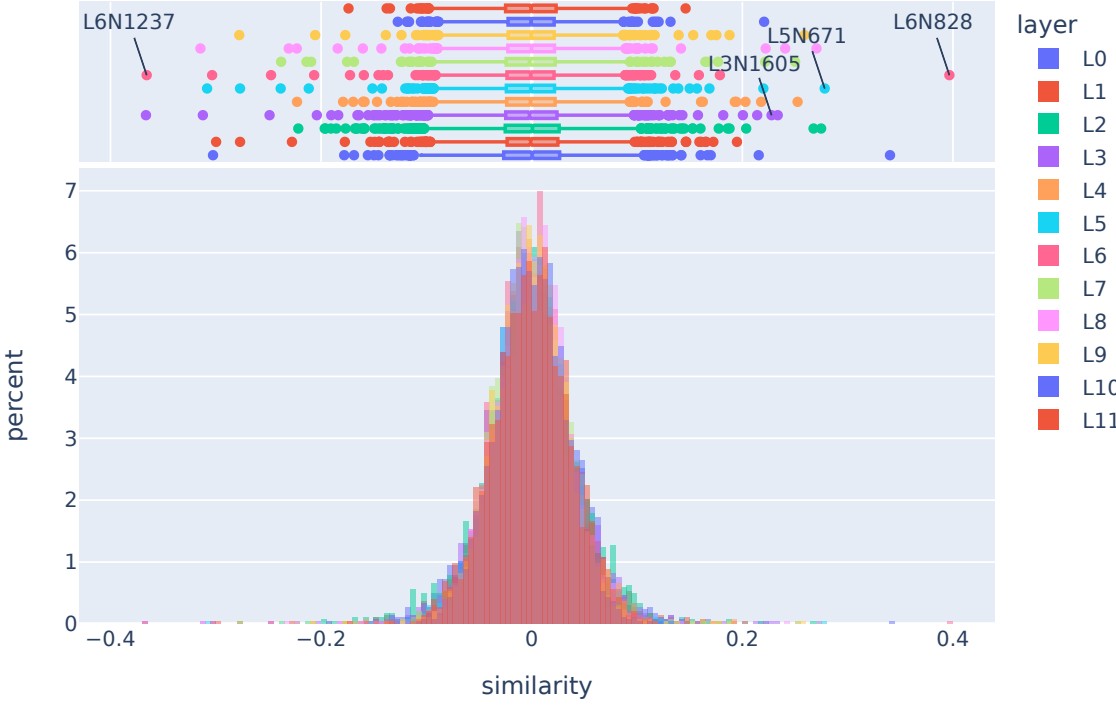

Figure A.16: Cosine similarity of neuron out-directions and the sentiment direction in GPT2-small

**directional ablation** A form of ablation experiment in which restrict the intervention to a single dimension. That is, assuming mean ablation, for dimension $d$ and prompt index $i$ out of $n$, we replace the residual stream vector $r_i$ with $r_i - r_i \cdot d + \sum_j \frac{r_j \cdot d}{n}$. (See Section 2.2)

**directional activation patching** A variant of activation patching introduced in this paper where we only patch a single dimension from a counterfactual activation. That is, for prompts $x_{\text{orig}}$ and $x_{\text{new}}$, direction $d$, a set of model components $\mathbb{C}$, we run a forward pass on $x_{\text{orig}}$ but for each component in $\mathbb{C}$, we patch/replace the output $o_{\text{orig}}$ with $o_{\text{orig}} - o_{\text{orig}} \cdot d + o_{\text{new}} \cdot d$. This is equivalent to activation patching a single neuron, but done in a rotated basis (where $d$ is the first column of the rotation matrix). (See Section 2.2)

**directional patching** See directional activation patching.

**freezing** When performing activation patching experiments, we sometimes choose to avoid patching a subset of model components with their activations from the flipped prompt, instead freezing the activations to their initial value from the forward pass on the original prompt. (See Section 2.2)

**froze** See freezing

**frozen attention** A type of freezing where the attention pattern is frozen from the original run so that the model still weights the value vectors in the same way, helping to isolate V-composition. (See Section 2.2)

**linear representation hypothesis** The idea that high-level concepts or "features" are represented linearly as directions in some representation space (Mikolov et al., 2013; Elhage et al., 2022; Park et al., 2023; Jiang et al., 2024).

**logit difference** The difference between the logits given to a particular pair of completions. To reduce noise, we can generalize this to the *average* difference between two sets of completions. In our case, the dichotomy of completions generally represent positive vs. negative sentiment. (See Section 2.2)

**logit difference metric** An evaluation metric, often used as the objective function by DAS and reported when activation patching, where we normalize the change in logit difference induced by patching such that 0 is no change and 1 corresponds to a sign change in the logit difference with no change in magnitude. (See Section 2.2)

**logit flip** An evaluation metric, ofted used in activation patching, which reports the percentage of examples where the prediction is flipped, i.e. the sign of the logit difference is flipped. For a single example, this is a binary value. (See Section 2.2)

**mean ablation** A type of ablation method, where we seek to eliminate the contribution of a particular component to demonstrate its importance, where we replace a particular set of activations with their mean over an appropriate dataset. (See Section 2.2)

**patching metric** A summary statistic used to quantify the results of an activation patching experiment. By default here we use the percentage change in logit difference as in Wang et al. (2022). (See Section 2.2)

**path patching** A variant of activation patching introduced in Wang et al. (2022) in which only the activations related to the residual stream paths between two sets of endpoints (senders and receivers) are patched, but the remainder of the network upstream of the receivers is frozen. Given a set $R$ of receivers, a sender attention head $h$, and paths $P$ between $h$ and each of $R$, activations from the mirrored dataset are patched into $P$ while keeping the remainder of the network fixed (aside from everything downstream of $R$). (See Section 2.2)

**sentiment activation** The projection of the residual stream at a given token position and layer onto the sentiment direction. (See the introduction to Section 3)

**sentiment direction** The direction in the residual stream space associated with the sentiment feature. (See the introduction to Section 3)

**sentiment summarizer** An attention head which is a critical component of a sentiment-driven task and acts via V-composition, writing information to an intermediate token position which is later read by a direct effect head.

**SST** Stanford Sentiment Treebank is a labelled sentiment dataset from Socher et al. (2013) described in Section 2.1.

**summarization motif** The phenomenon where sentiment is not solely represented on emotionally charged words, but is additionally summarised at intermediate positions without inherent sentiment, such as punctuation and names.

**V-composition** When the value vectors of a downstream head contain information written by the output of an upstream attention head (Elhage et al., 2021b).

**value-weighted attention** The attention pattern weighted by the norm of the value vector at each position as per Kobayashi et al. (2020). We favor this over the raw attention pattern as it filters for *significant* information being moved.

**zero ablation** A type of ablation method, where we seek to eliminate the contribution of a particular component to demonstrate its importance, where we replace a particular set of activations with their mean over an appropriate dataset. (See Section 2.2)