# OpenReview forum: "Language Models Linearly Represent Sentiment"
_EMNLP/2024/Workshop/BlackBoxNLP — BlackboxNLP 2024_

### Official Review · Reviewer_KKaL · 2024-09-08

**Overall Assessment:** 4
**Confidence:** 3

**Best Paper:**

1

**Best Paper Justification:**

N/A (why is this field mandatory?)

**Comments Questions Suggestions And Typos:**

The editing, especially of the second half of the paper, leaves much to be desired. Figures are relying on color for differentiation (can't be understood in black-and-white printouts or by colorblind folks). Figure 3 should be a Table and follow Figure 4, not preced it. A citation is garbled in l.361, another is written wrong in l.433. Formatting of paragraph headers is completely inconsistent. Table captions should be below the tables. The information in Tables 2--4 is indecipherable and probably shouldn't be presented in table form. A closing paren is missing in l.511.

Please don't call your own results "remarkable" (l.411) or "tantalizing" (l.583). It's unseemly.

**Paper Summary:**

The paper presents an analysis of LLMs' performance on sentiment analysis tasks through search for summarization motifs in toy datasets followed by applying them as ablations in real-world datasets. The findings are nontrivial and interesting, particularly of note is that where "intermediate" levels store sentiment information in non-semantic tokens such as commas and periods, as if they were placeholders for further processing levels.

**Summary Of Strengths:**

* An incremental analysis method which brings interesting results
* Novel findings in the space of sentiment analysis interpretability
* Extensive experimentation

**Summary Of Weaknesses:**

* The overall story becoms much less coherent starting from section 4, becoming a laundry list of results without a common thread put forward in writing.

---

### Official Review · Reviewer_e2tE · 2024-09-09

**Overall Assessment:** 3
**Confidence:** 3

**Best Paper:**

1

**Best Paper Justification:**

N/A

**Comments Questions Suggestions And Typos:**

None

**Paper Summary:**

The paper discusses sentiment representation in a series of decoder models. With a series of correlational and causal representational experiments, the authors discover linear representation of sentiment in the models of interest. Additionally, they show something they call 'summarisation motif' -- sentiment information stored in representations of delimiting tokens that are not inherently sentiment-related (for instance, punctuation marks).

**Summary Of Strengths:**

The experiments are interesting and designed well. The triangulation of 'sentiment dimension' using different methods and datasets -- especially, extrapolating from findings on toy datasets to more realistic data -- is convincing. I definitely learned something interesting from the paper.

**Summary Of Weaknesses:**

I have concerns about some aspects of presentation, mostly. But some of them have to do with content as well.

1. The connection between alleged world models in the representations of LMs and sentiment is not immediately clear. Why discuss them together at all? Sentiment is not part of world model at least in the conventional truth-conditional sense of the latter.
2. The suggestion that LMs have track entities and states and have a world model in that sense should not be taken as a proven fact, see for instance Kim and Schuster (2023) for a critique of relevant results.
3. Similarly, I think a caveat is due somewhere around the discussion about stuff that's linearly representable. It seems almost like the authors suggest that for practically anything a linear representation can be found. But even for well-established dimensions things might be a bit more complex. I refer here to discussion in Elazar et al. (2021) and Ravfogel et al. (2022) about 'recoverability' of linearly removed information (although this might be due to removal method artefacts).
4. 'Summarization motif' is maybe not the best name for a phenomenon if it forces you to have a footnote just to say that it should not be confused with a completely different phenomenon.
5. The part about negation doesn't look like it's motivated by the logic of the paper. Why is it here? What does it show? I was a bit lost.

REFERENCES


Kim, N. and Schuster, S. (2023). Entity tracking in language models. Proceedings of ACL (pp. 3835–3855).
Elazar, Y., Ravfogel, S., Jacovi, A., and Goldberg, Y. (2021). Amnesic probing: Behavioral explanation with amnesic counterfactuals. TACL, 9:160–175.
Ravfogel, S., Vargas, F., Goldberg, Y., and Cotterell, R. (2022). Adversarial concept erasure in kernel space. Proceedings of EMNLP (pp. 6034–6055).

---

### Official Review · Reviewer_oYwd · 2024-09-12

**Overall Assessment:** 4
**Confidence:** 2

**Best Paper:**

2

**Best Paper Justification:**

The findings are interesting and novel (particularly the motifs), and the investigation is quite rigorous

**Comments Questions Suggestions And Typos:**

- The idea of logit difference over sets is not clear. The way it is presented here suggests that each positive word has a corresponding word in the negative set, but the index of a word is then important and it is unclear how this index is determined. The other way might be that the pairwise logit difference for each pair of words (one from the positive and one from the negative set) is computed, but that doesn't seem to be what is described

**Paper Summary:**

The paper presents an analysis technique to identify a "sentiment direction" in language model hidden representations. The paper finds that sentiment information is largely encoded linearly, supporting this finding with a variety of experimental evidence, including finding similar directions through multiple estimation methods, generalization of a sentiment direction determined on synthetic data to a naturalistic dataset, and causal manipulations of representations using this information. The paper also finds "summarization motifs", the incidence of predictive information about sentiment not being localized in just semantically relevant ("sentiment bearing") words, but also in the representation of a symbol like a comma. This localization is also found to generalize to other settings, and causally affect model predictions (as confirmed through ablations).

**Summary Of Strengths:**

- The paper presents a thorough investigation of the question of whether sentiment is linearly encoded in language model representations
- There are interesting and unexpected novel findings discussed like the idea of summarization motifs
- The experimentation is rigorous, with carefully designed controls

**Summary Of Weaknesses:**

- The writing is a little dense, and includes a lot of novel terminology (for someone who might not be very familiar with work in probing/interpretability), but the detailed glossary does help make it more accessible

---

### Decision · Program_Chairs · 2024-09-19

**Decision:**

Accept

**Comment:**

Reviewers agree that the empirical contributions of this work are well-executed and interesting. This paper is also a great fit for this workshop. I vote to accept.

However, In the camera-ready, it would be great to address some significant points raised by reviewers. Reviewer e2tE points out issues in the framing of the paper—namely, why world models are used as a framing device (with which I also take issue), and certain potentially over-optimistic assumptions about what LMs are currently capable of. Reviewers e2tE and KKaL also agree that the quality of the writing and presentation becomes significantly lower-quality in the later sections of the paper, and that there appear to be many disconnected results and topics that could be either (i) better linked by the writing, or (ii) moved to an appendix or separate paper.